# The therapeutic potential of multiclonal tumoricidal T cells derived from tumor infiltrating lymphocyte-derived iPS cells

Takeshi Ito[1,2], Yohei Kawai[1], Yutaka Yasui[1,3], Shoichi Iriguchi[1], Atsutaka Minagawa[1], Tomoko Ishii[1], Hiroyuki Miyoshi [4], M. Mark Taketo[4], Kenji Kawada [2], Kazutaka Obama[2], Yoshiharu Sakai[5] & Shin Kaneko [1✉]

Tumor-infiltrating lymphocytes (TIL), which include tumor-specific T lymphocytes with frequency, are used for adoptive cell transfer therapy (ACT) in clinical practice. The optimization of TIL preparation has been investigated to reduce the senescence and increase the abundance of TIL, as both the quality and quantity of the transferred cells have great influence on the outcome of TIL-based ACT (TIL-ACT). Considering the effects of cell reprogramming on senescence, we expected that the anti-tumor effect could be enhanced by TIL regeneration. To confirm this hypothesis, we established tumor-specific TIL-derived iPS cells (TIL-iPSC) with human colorectal cancer specimens. T cells differentiated from TIL-iPSC (TIL-iPS-T) retained not only intrinsic T cell functions and tumor specificity, but also exhibited improved proliferation capacity and additional killing activity. Moreover, less differentiated profiles and prolonged persistency were seen in TIL-iPS-T compared with primary cells. Our findings imply that iPSC technology has great potential for TIL-ACT.

[1] Shin Kaneko Laboratory, Department of Cell Growth and Differentiation, Center for iPS Cell Research and Application (CiRA), Kyoto University, Shogoin-Kawahara-cho, Sakyo-ku, Kyoto, Japan. [2] Department of Surgery, Graduate School of Medicine, Kyoto University, Shogoin-Kawahara-cho, Sakyo-ku, Kyoto, Japan. [3] Thyas Co. Ltd., Yoshida-Shimo-Adachi-cho, Sakyo-ku, Kyoto, Japan. [4] Institute for Advancement of Clinical and Translational Science (iACT), Kyoto University, Yoshida-Honmachi, Sakyo-ku, Kyoto, Japan. [5] Osaka Red Cross Hospital, Fudegasaki-cho, Tennoji-ku, Osaka, Japan. ✉email: kaneko.shin@cira.kyoto-u.ac.jp

Tumor-infiltrating lymphocytes (TIL) are a polyclonal population that contains tumor-specific T cells with high frequency. These tumor-specific T cells mainly recognize neoantigens resulting from somatic mutations in cancer cells. For this reason, the mutation profiles are important to determine the efficacy of TIL-based cancer immunotherapy. Immunotherapies tend to be effective against malignancies with high mutation burden such as melanoma, non-small cell lung cancer, DNA mismatch repair-deficient (MMR-D)/microsatellite instability-high (MSI-H) cancers and so on[1–7]. While immune checkpoint inhibitors play a leading part in current immunotherapies, cell-based strategies offer promising results as well.

TIL themselves are considered an attractive cell resource for immunotherapies due to their aforementioned characteristics. The Rosenberg lab has established ex vivo TIL culture methods to initiate TIL-based adoptive cell transfer therapy (TIL-ACT) against cancers, mainly metastatic melanoma[8–11]. Owing to multiple TIL optimization and therapeutic protocols, this form of personalized immunotherapy has produced good results in the treatment of metastatic melanoma. The objective response rates reported from clinical trials were 40–72% and contained a number of durable complete remission cases[7,12–19]. Nowadays, the target of TIL-ACT is not restricted to melanoma but has expanded to various kinds of epithelial cancers. Despite limited clinical data, several successful cases have been reported[20–23].

Feedback from the bedside has provided predictive factors regarding the therapeutic effects of TIL-ACT. The factors associated with transferred TIL are the high number of transferred cells[16,24], the high number or fraction of transferred CD8+ cells[16,17,25], longer persistency in peripheral blood[15,26,27], longer telomeres[13,15,28], and less differentiated phenotypes[15,16]. Thus, preparing a significant number of juvenile tumor-specific CD8+ cells leads to better clinical outcomes, and refinements to TIL-ACT have focused on preparing young and less differentiated TIL to increase the effectiveness[14,27,29,30]. However, it is practically difficult to prepare ideal cells, because the main populations in TIL are often senescent and fairly differentiated. In response, induced pluripotent stem cell (iPSC) technology is being considered.

iPSC are pluripotent stem cells generated from somatic cells using exogenous reprogramming factors. The generation of iPSC was initially reported with murine and human fibroblasts[31,32], but various somatic cells can be reprogrammed. Among them, T cells are an interesting cell source for iPSC generation, because the TCR rearrangement pattern of the parent T cell can be passed to the T cell-derived iPS (T-iPSC)[33–36]. Moreover, T cells regenerated from T-iPSC retain the same TCR information and antigen specificity. We and other groups reported the proof of concept by regenerating antigen-specific CD8+ T cells[37,38]. The regenerated T cells showed restored T-cell function and a rejuvenated profile[38]. Furthermore, iPSC technology enables an unlimited supply of antigen-specific CD8+ T cells. These advantages regarding T-cell regeneration from T-iPSC have implications for conventional TIL-ACT.

Based on the above, we hypothesized that reprogramming could overcome the problems of conventional TIL-ACT by providing regenerated TIL with a rejuvenated profile and more T-cell function. In addition, the TIL regeneration is theoretically unlimited. In this study, we established tumor-specific TIL-derived iPS cells (TIL-iPSC) for human colorectal cancers containing both DNA mismatch repair profiles: deficient (MMR-D) and proficient (MMR-P). We then regenerated T cells from the TIL-iPSC (TIL-iPS-T). We also performed multiclonal T-cell regeneration, which, unlike chimeric antigen receptor (CAR) or T-cell receptor (TCR) transduced immune cells directed to a particular target, prevents cancer cells from escaping the T-cell

immunity. To verify the quality of TIL-iPS-T from the T-cell perspective, we compared TIL-iPS-T with primary TIL in vitro. In addition, we investigated the therapeutic potential of TIL-iPS-T in patient-derived spheroids xenograft (PDSX) mice.

## Results

**Collecting and expanding TIL from human colorectal cancer specimens.** Clinical samples were obtained from 16 colorectal cancer patients who underwent surgical resection of primary tumors (Supplementary Table 1). TIL and cancer epithelium were separated from primary tumor specimens and later processed (Fig. 1a). From the cancer epithelium and normal mucosal tissue adjacent to the primary tumor site, autologous spheroid lines were established[39,40]. Profiles of the collected TIL were evaluated prior to cytokine-based expansion (Fig. 1b and Supplementary Fig. 1a, b). The CD3+ fraction in the mononuclear cell fraction of TIL was significantly lower than in peripheral blood mononuclear cells (PBMC). The fraction of cytotoxic T lymphocytes (CTL; defined as CD3+CD4−CD8+) was significantly lower in TIL than in PBMC. Consistent with previous reports, tumor-infiltrating CTL (TI-CTL) frequently expressed PD-1 and were concentrated in the effector memory (CD45RA−CD62L−) T-cell population[41–46]. The number of TI-CTL contained in an ~5 mm³ tumor block was quite low (Fig. 1c). Thus, TI-CTL expansion was needed before the tumor-specific TI-CTL selection phase.

The TIL expansion protocol with a high dose of IL-2 is well established and applied to various kinds of malignancies[8,9,11,41–43,47]. To improve the expansion efficiency and expanded TIL quality, researchers have tried modifying the culture conditions by adding other cytokines or compounds[30,46,48–50]. We initiated TIL expansion with different doses of IL-2-based cytokine cocktails. The addition of more cytokines improved the TI-CTL expansion efficiency (Fig. 1d). The addition also upregulated CD62L and increased central memory (CD45RA−CD62L+) T-cell populations (Supplementary Fig. 1c). Because we were concerned about the diversity loss of TI-CTL during the cytokine expansion process, the TCR Vβ-based repertoire was compared between pre- and post-cytokine expansion TI-CTL. However, no diversity loss was observed in cytokine-expanded TI-CTL (Fig. 1e).

**Selection of tumor-specific TI-CTL from cytokine-expanded bulk populations.** Though TIL contain more tumor-specific CTL than peripheral blood, bystander CTL are actually the main population of TI-CTL[51,52]. Accordingly, technologies regarding tumor-specific CTL selection are required. Several surface markers related to T-cell activation, exhaustion, and tissue residency are thought to be useful for selecting tumor-specific CTL[44,51,53,54]. However, these reported strategies are available only for unbiased, pre-expanded TI-CTL. We, therefore, attempted another strategy in which autologous cancer spheroids were employed for the tumor-specific CTL selection (Fig. 2a). We observed the upregulation of either CD107a or 4-1BB (CD137), two markers to verify T-cell reactivity against antigens[55,56], on TI-CTL after co-culture with autologous cancer spheroids (Fig. 2b). Reactive populations against cancer spheroids were sorted and expanded by a phytohemagglutinin-P (PHA)/PBMC expansion protocol. The expansion method corresponds to the rapid expansion protocol largely applied in ACT therapy[11] and enables the obtainment of plenty of tumor-reactive TI-CTL.

Though the enrichment of tumor-reactive TI-CTL after the expansion was confirmed, the activation was partially inhibited by an HLA class I blocking antibody (Fig. 2c). This result suggested that tumor nonspecific TI-CTL clones, i.e., cells

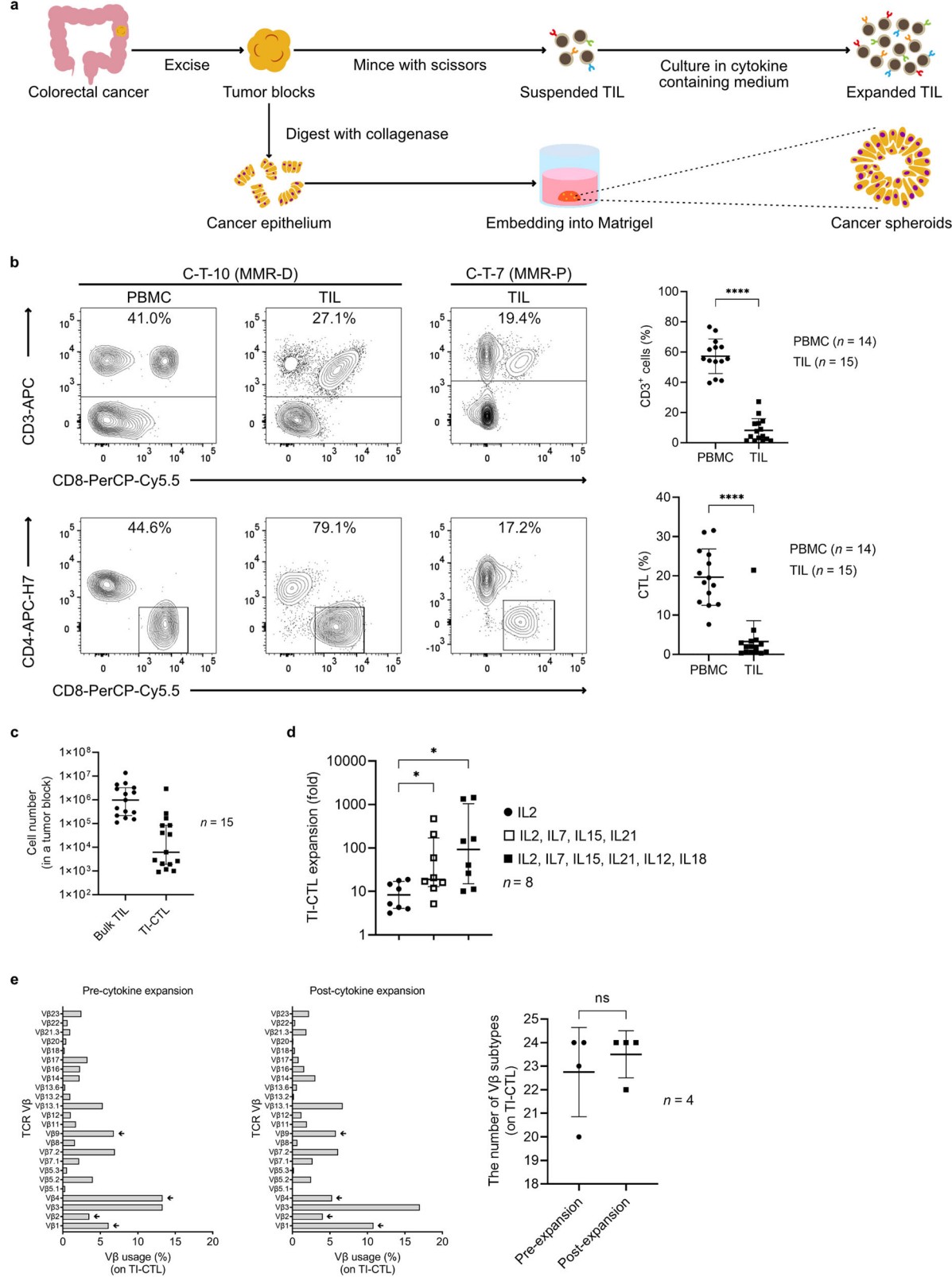

whose activation against cancer spheroids was independent of TCR-HLA class I interaction, could be picked up. Moreover, the repetitive bulk expansion resulted in narrowing the diversity of TI-CTL (Supplementary Fig. 2). According to the above reasons, we decided to pick up tumor-specific clones from the bulk tumor-reactive population to realize the multiclonal TIL regeneration.

Thereafter, we evaluated the activation of TI-CTL against the cancer spheroids on a per Vβ-subtype basis. The inhibitory effects of HLA class I blocking antibody on TI-CTL activation induced by co-culture with cancer spheroids were different in each Vβ-based subgroup, as expected (Fig. 2d and Supplementary Fig. 3a, b). The Vβ subgroups whose activation against cancer spheroids were efficiently inhibited with HLA class I blocking antibody were

**Fig. 1 Collecting and expanding TIL from human colorectal cancer specimens. a** Processing workflow of clinical specimens. **b** The T-cell population in PBMC and TIL. Left: representative flow cytometry plots of PBMC and TIL. Upper data were gated on live cells. Lower data were gated on CD3$^+$ live cells. Right: quantification of CD3$^+$ cells or CTL frequency in PBMC and TIL. ****$P < 0.0001$, two-tailed unpaired $t$ test. Dots represent data from individual cases. Means are shown; error bars represent SD. **c** Number of bulk TIL and TI-CTL from an ~5 mm$^3$ tumor block. Dots represent data from individual cases. Medians are shown; error bars represent interquartile ranges. **d** Expansion data of TI-CTL with cytokine cocktails. Expansion folds of TI-CTL were compared on day 14. *$P < 0.05$, paired nonparametric analysis (Friedman test followed by Dunn's multiple comparisons test). Dots represent data from eight individual cases. Medians are shown; error bars represent interquartile ranges. **e** The TCR Vβ repertoire analysis of TI-CTL based on flow cytometry. Left: representative clonogram from both pre- and post-cytokine expansion TI-CTL for C-T-10 MMR-D. Arrows indicate Vβ subtypes containing tumor-specific populations. Right: dots indicate the number of detected Vβ subtypes from individual cases. NS, not significant, two-tailed paired $t$ test. Means are shown; error bars represent SD.

regarded as tumor-specific populations and selectively expanded again. These tumor-specific populations were rarely activated by non-cancer spheroids established from normal mucosal tissue (Supplementary Fig. 3c). The tumor-specific TI-CTL showed killing activity only against cancer spheroids and not against non-cancer spheroids (Fig. 2e).

**Establishment of tumor-specific TIL-iPSC and regeneration of TIL-iPS-T.** Some groups including ours have reported antigen-specific T-cell regeneration via iPSC technology[37,38,57–59]. The establishment of TIL-iPSC in human melanoma cases has also been reported[60]. Nevertheless, there are no reports on antigen-specific T-cell regeneration from TIL. Therefore, we tried to regenerate CTL via TIL-iPSC that were established from tumor-specific TI-CTL (Fig. 3a). We reprogrammed the selected TI-CTL with Sendai virus vectors by modifying previous protocols[36,60]. A sufficient number of colonies was acquired from all selected TI-CTL, and we could regenerate TIL-iPS-T clones except C-T-7 Vβ22 (Supplementary Table 2). For the T cell regeneration, we applied feeder-free (FF) protocols[61,62] composed of two phases. The first phase is hematopoietic progenitor cell (HPC) induction through the FF embryoid body method. In the second phase, HPC were transferred onto FcDLL4-coated plates and committed to T-cell lineage. Indeed, some CD4$^-$CD8$^+$ single positive (SP) populations could be obtained during the FcDLL4 culture, but residual CD4$^+$CD8$^+$ double-positive (DP) populations need to be matured by stimulating with functional CD3 antibody. After the maturation step, the DP populations turned into SP populations. The expression level of CD5 diminished during the maturation step, but regenerated TIL-iPS-T stably expressed other kinds of CTL-related molecules including CD8αβ heterodimers, CD3, TCRαβ, and CD7 (Fig. 3b and Supplementary Fig. 4a). Of note, TIL-iPS-T upregulated several phenotype-related markers, especially CD62L, CD28, and *TCF-1* (*TCF-7*), unlike parent TI-CTL (Fig. 3c, d). Besides the markers, an enlargement of mitochondria biomass was found in TIL-iPS-T (Fig. 3e). Regarding exhaustion-related markers, TIL-iPS-T mildly expressed PD-1 but not other markers (Supplementary Fig. 4b).

We previously reported that T cells regenerated from T-iPSC could lose their antigen specificity due to additional TCR rearrangements during the T-cell induction phase[59]. Thus, we checked the TCR rearrangement patterns in TIL-iPS-T that did not experience any selection regarding tumor specificity during the T-cell regeneration process. Though half of the TIL-iPS-T clones showed dual TCRα patterns, these patterns seemed to be endowed from the original clones, and no additional rearrangement was found in either the TCRα or TCRβ chains (Fig. 4a, b).

We considered whether the absence of additional TCR rearrangements arose from differences in the T-cell induction protocols. To assess this hypothesis, we performed T-cell inductions from two T-iPSC clones, TKT3V1–7 and GPC3 16-

1, by applying on-feeder (OF) and FF protocols (Supplementary Fig. 5a). Because TCR rearrangement is triggered with recombination activating gene (*RAG*) during the DP stage, we focused on *RAG1* and *RAG2* expressions in DP cells from both protocols but found no obvious difference (Supplementary Fig. 5b).

Even without an explanation for the no additional TCR rearrangements in TIL-iPS-T, we evaluated the tumor reactivity against autologous spheroids. TIL-iPS-T derived from tumor-specific TI-CTL were well activated by co-culturing with cancer spheroids (Fig. 4c). On the other hand, TIL-iPS-T from tumor nonspecific TI-CTL, C-T-7 Vβ5.3, did not respond to cancer spheroids. These results are consistent with no additional TCR rearrangements during T-cell differentiation.

**T cell-related function profiles of regenerated TIL-iPS-T.** After expanding the matured TIL-iPS-T, we characterized the cells by comparing them with primary TI-CTL. First, we checked their proliferation capacities in response to PHA/PBMC expansion, which corresponds to the rapid expansion protocol described above. TIL-iPS-T showed significantly higher proliferation capacity than TI-CTL in all of the compared clones (Fig. 5a). Although some clones in TI-CTL, especially C-T-10 Vβ4 and C-T-11 Vβ5.1, were difficult to expand, TIL-iPS-T showed steady expansion in all examined clones. Next, we measured cytokine-producing ability. Both TIL-iPS-T and TI-CTL slightly produced interferon-γ (INF-γ) and tumor necrosis factor (TNF), but hardly IL-2, in response to co-culturing with cancer spheroids (Fig. 5b and Supplementary Fig. 6a). However, improved IL-2 production was found in TIL-iPS-T when stimulated with phorbol myristate acetate (PMA) plus ionomycin (ION) (Fig. 5b and Supplementary Fig. 6b). The cytokine production pattern with PMA plus ION confirmed that TIL-iPS-T was in a less differentiated state than TI-CTL (Supplementary Fig. 6c).

Finally, we evaluated the killing ability of TIL-iPS-T against autologous spheroids. TIL-iPS-T from tumor-specific clones demonstrated their killing activity only against cancer spheroids (Fig. 5c, d). While the killing activity levels of TIL-iPS-T were similar to TI-CTL for MMR-D, increased killing activity was found in TIL-iPS-T for MMR-P. TIL-iPS-T from tumor nonspecific clones also exhibited killing activity against cancer spheroids to some extent (Fig. 5e). We speculated the additional killing of TIL-iPS-T was due to NK cell-like behavior, based on the character of regenerated T cells[58,63]. To address this point, we blocked TCR-HLA class I interaction during the killing assay. The blocking confirmed that TIL-iPS-T utilized both TCR-dependent and -independent killing activities, unlike TI-CTL (Supplementary Fig. 7a, b). Moreover, TIL-iPS-T expressed a higher level of NK cell-related markers, especially CD336 (NKp44) and CD337 (NKp30), which are involved in MHC non-restricted natural cytotoxicity, compared with TI-CTL (Supplementary Fig. 7c).

Because TIL-iPS-T demonstrated promising results in killing assays that utilized dissociated target cells, we next observed whether TIL-iPS-T could destroy 3D structured cancer spheroids.

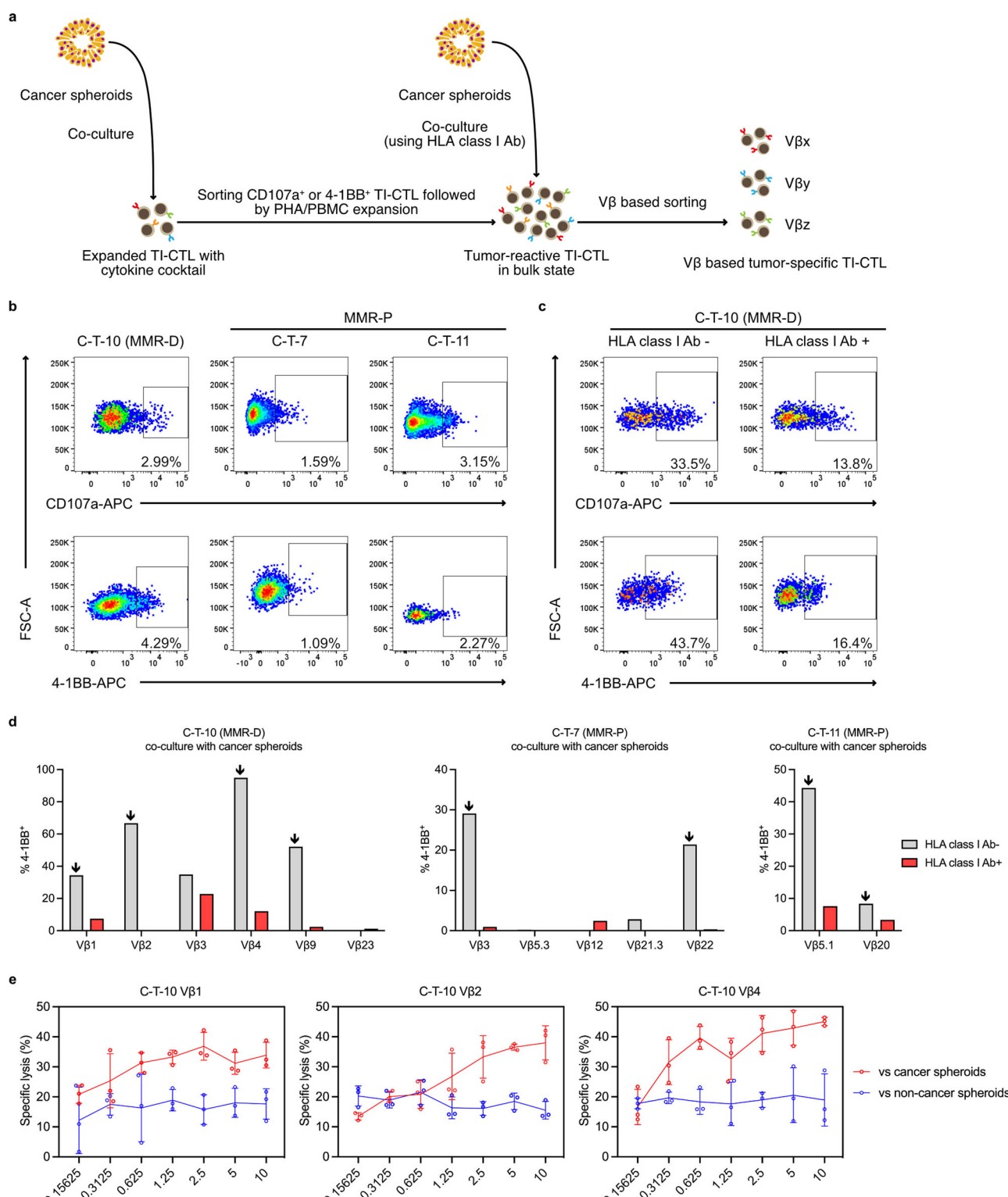

**Fig. 2 Selection of tumor-specific TI-CTL from cytokine-expanded populations. a** Overview of the tumor-specific TI-CTL selection from cytokine-expanded populations. **b** Cytokine-expanded TI-CTL were co-cultured with autologous cancer spheroids. CD107a+ or 4-1BB+ populations were sorted and selectively expanded by the PHA/PBMC method. All data were gated on CD45+ live cells. **c** Tumor-reactive TI-CTL were co-cultured with cancer spheroids again with or without HLA class I antibody. Representative data of three independent experiments are shown from C-T-10 MMR-D. All data were gated on CD45+ live cells. **d** Activation of tumor-reactive TI-CTL against cancer spheroids was evaluated on a per Vβ-subtype basis. HLA class I blocking antibody was used for the prediction of TCR-HLA class I axis dependency for the TI-CTL activation. The arrows designate Vβ subtypes that mainly depend on TCR-HLA class I interactions for their activation. Left: data from MMR-D. Representative data of two independent experiments are shown. Middle and right: data from MMR-P. Representative data of three independent experiments for each are shown. **e** Killing function data of the selected Vβ-based TI-CTL subgroups against autologous spheroids. Representative data of three independent experiments are shown. The $^{51}$Cr release assay was applied for the quantification. Dots represent individual values. Means are connected by lines; error bars represent SD. $n = 3$ per point.

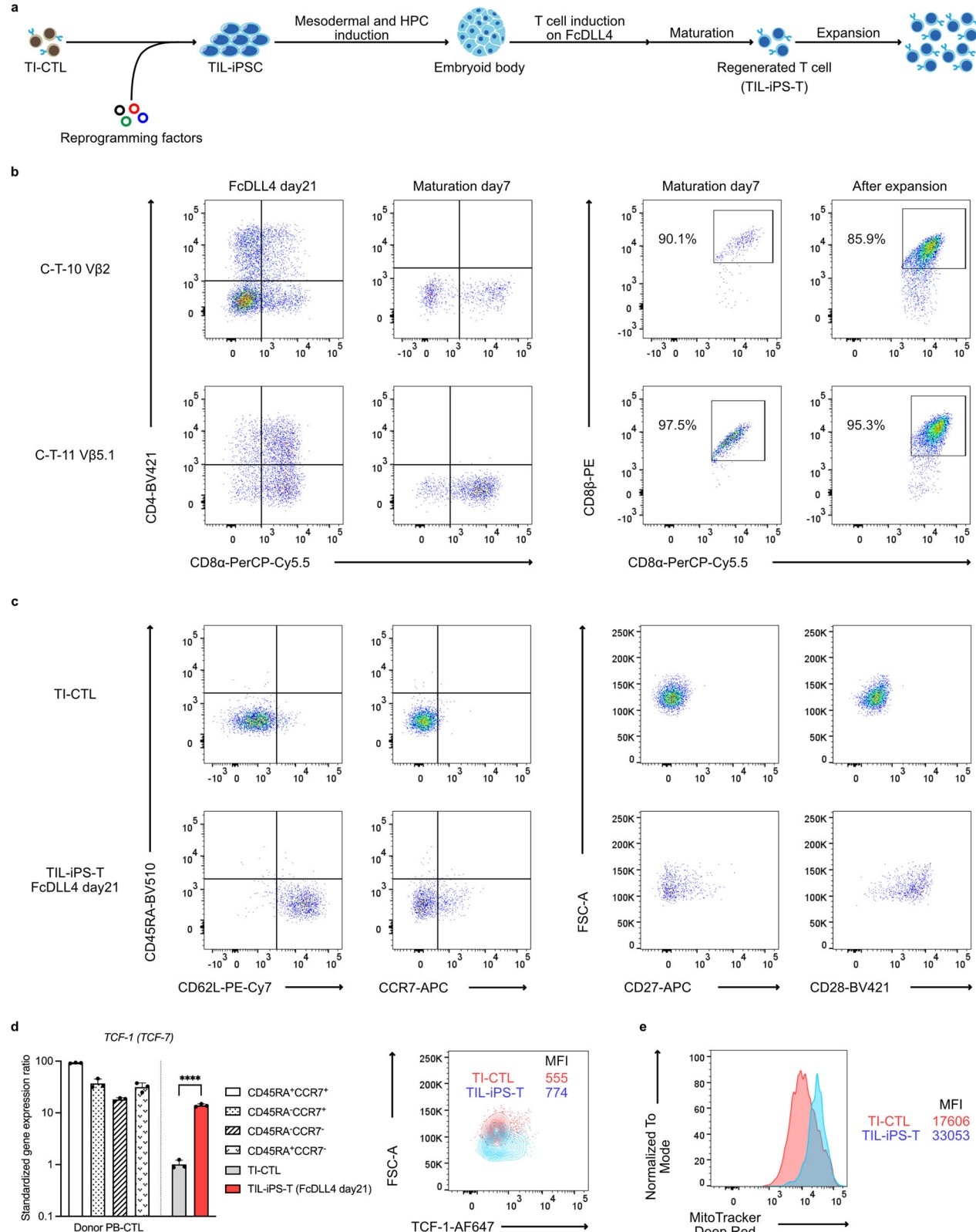

Time-lapse imaging revealed that TIL-iPS-T from C-T-10 Vβ4 collapsed the 3D-structured cancer spheroids (Supplementary Movie 1).

Taken together, TIL-iPS-T showed improved proliferative functions, cytokine production patterns consistent with a less differentiated phenotype, and additional cytotoxicity via a TCR-independent manner. According to the above results, we concluded that T-cell regeneration brings positive effects on TIL-iPS-T with regard to T cell-related functions.

**Persistency-related profiles of regenerated TIL-iPS-T.** Some parameters predicting the efficacy of ACT have been reported, especially the persistency of the transferred T cells[26]. Therefore, we evaluated persistency-related parameters for

**Fig. 3 T-cell regeneration from tumor-specific TI-CTL-derived iPSC. a** Overview of the TIL-iPSC establishment and T-cell regeneration. TIL-iPSC were established from tumor-specific TI-CTL. Sendai virus was used for the reprogramming. Feeder-free protocols were applied for the T-cell regeneration from TIL-iPSC. HPC hematopoietic progenitor cell. **b** Flow cytometry analysis regarding T-cell lineage markers at each differentiation stage. Left panels were gated on CD45$^+$ live cells. Right panels were gated on CD45$^+$CD4$^-$CD8$\alpha^+$ live cells. **c** Phenotype-related markers on TIL-iPS-T. Representative data of four independent experiments. Data were acquired from C-T-11 V$\beta$5.1 clone. All data were gated on CD45$^+$CD4$^-$CD8$\alpha^+$CD8$\beta^+$ live cells. **d** TCF-1 (TCF-7) expression on TIL-iPS-T. TI-CTL and TIL-iPS-T data were acquired from C-T-11 V$\beta$5.1 clone. PB-CTL (peripheral blood-derived CTL) were from a healthy donor. Left: quantification by qPCR. mRNA was extracted from the CD45$^+$CD4$^-$CD8$\alpha^+$CD8$\beta^+$ live cell population. Representative data of two independent experiments are shown. The expression levels were standardized with the $\Delta$CT method by ACTB expression and further standardized with the $\Delta\Delta$CT method by the $\Delta$CT mean of TCF-1 expression on TI-CTL. ****$P < 0.0001$, two-tailed unpaired $t$ test. Dots represent individual data. Means are shown; error bars represent SD. $n = 3$ per sample. Right: Representative flow cytometry data of two independent experiments are shown. Data were gated on CD45$^+$CD4$^-$CD8$\alpha^+$CD8$\beta^+$ cells. MFI mean fluorescence intensity. **e** Mitochondria biomass on TIL-iPS-T. Data were acquired from C-T-11 V$\beta$5.1 clone. Histograms were gated on CD45$^+$CD4$^-$CD8$\alpha^+$CD8$\beta^+$ live cells. Representative data of four independent experiments are shown.

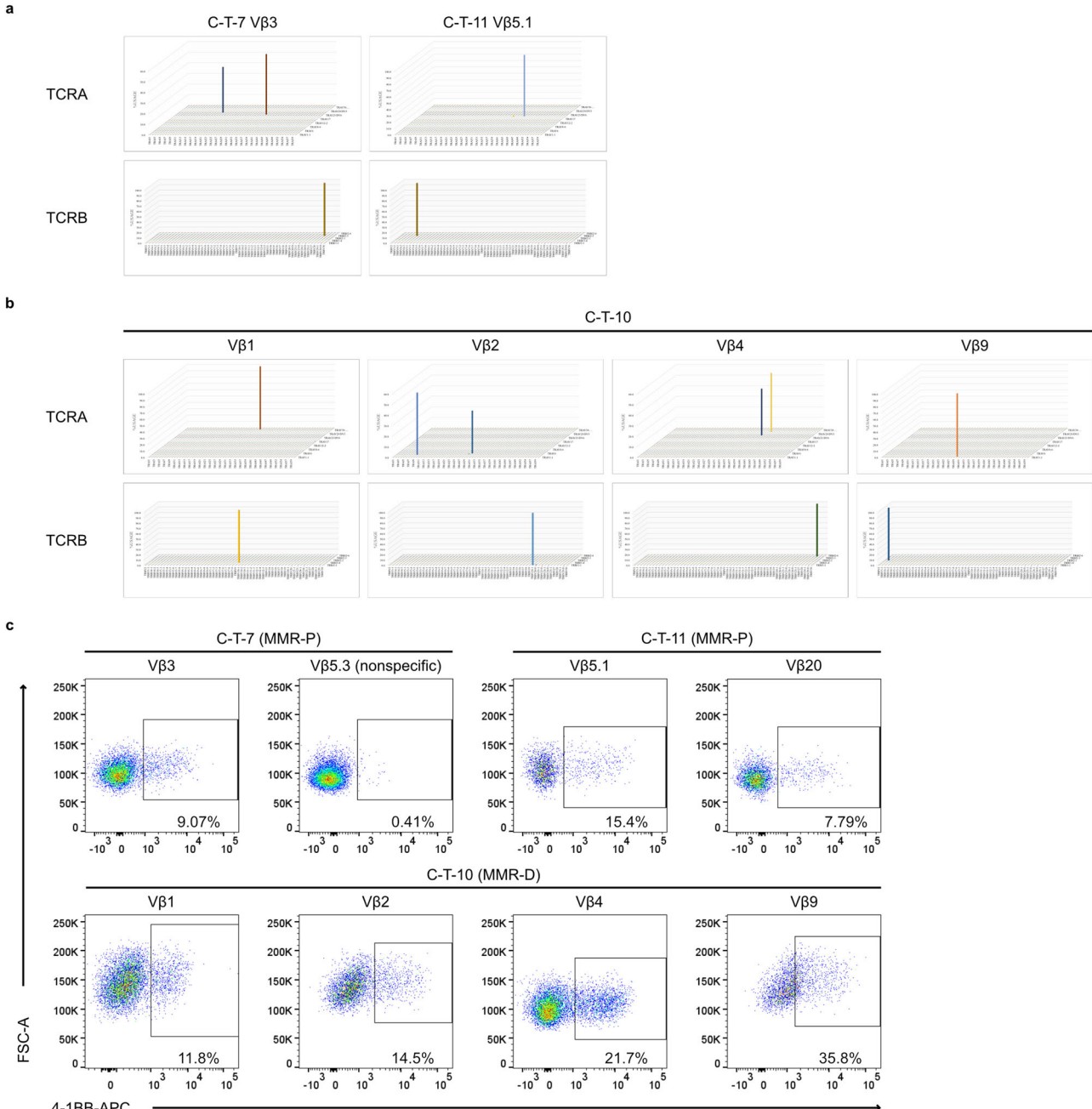

**Fig. 4 TIL-iPS-T retained original TCR information and tumor specificity. a**, **b** Repertoire analysis of TIL-iPS-T from MMR-P (**a**) and MMR-D (**b**). NGS was applied in the analysis. **c** Flow cytometry analysis regarding the tumor reactivity of TIL-iPS-T. Representative data of three independent experiments are shown. Data were gated on live cells.

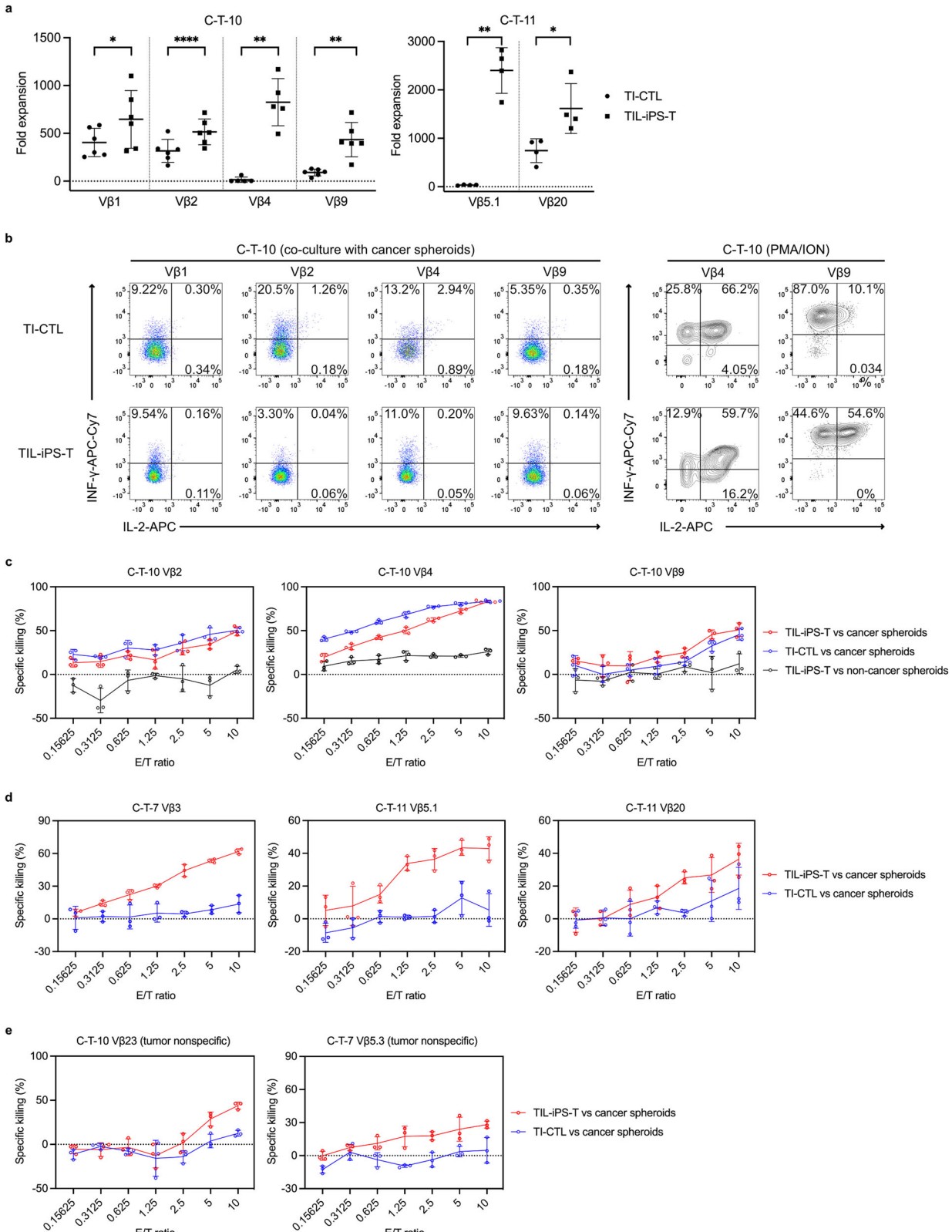

TIL-iPS-T and TI-CTL. Telomere length is one such parameter[28]. TIL-iPS-T showed a longer telomere length than TI-CTL and a similar one as peripheral blood-derived CTL for MMR-D (Fig. 6a). Mitochondrial functions are also crucial factors for cell viability and therapeutic effects[30,64]. Though TIL-iPS-T for MMR-D exhibited a higher spare respiratory capacity (SRC) than TI-CTL, we concluded this property was independent of T-cell regeneration and acquired by chance, because TIL-iPS-T for MMR-P had a lower SRC than TI-CTL (Fig. 6b).

Regarding the persistency of the transferred T cells within solid tumors, local proliferation and apoptosis sensitivity of the transferred cells in the tumor core are important after homing to the lesion and recognizing the cancer cells. We investigated

**Fig. 5 T cell-related function profiles of regenerated TIL-iPS-T. a** Proliferation capacity of TI-CTL and TIL-iPS-T reacting to the PHA/PBMC method. *$P < 0.05$, **$P < 0.01$, ****$P < 0.0001$, two-tailed paired $t$ test. Dots represent data from individual experiments. Means are shown; error bars represent SD. **b** Cytokine production capacity was evaluated for C-T-10 MMR-D. Representative data of three independent experiments are shown. Left: data were acquired by co-culturing with cancer spheroids and gated on CD45$^+$ cells. Right: data were acquired by stimulating with PMA plus ION and gated on CD45$^+$CD4$^-$CD8α$^+$CD8β$^+$ cells. **c, d** Killing function of tumor-specific clones was quantified by the luminescence-based killing assay. Data from MMR-D (**c**) and MMR-P (**d**). Representative data of four independent experiments are shown. Dots represent individual values. Means are connected by lines; error bars represent SD. $n = 3$ per point. **e** Killing function of tumor nonspecific clones was quantified by the luminescence-based killing assay. Representative data of four independent experiments are shown for C-T-10 Vβ23 and seven independent experiments are shown for C-T-7 Vβ5.3. Dots represent individual values. Means are connected by lines; error bars represent SD. $n = 3$ per point.

these parameters by co-culturing the cells with autologous cancer spheroids. No obvious difference was found in the CFSE-dilution assay (Fig. 6c), but TIL-iPS-T were more resistant to apoptosis than TI-CTL (Fig. 6d).

All together, these results indicate that T-cell regeneration brings about telomere elongation and increased apoptosis tolerance regarding cell persistency.

**Characterization of regenerated TIL-iPS-T in PDSX model.** We finally characterized TIL-iPS-T in vivo using the PDSX model. Based on the Winn assay, TIL-iPS-T inhibited the establishment of PDSX mice when the cells were subcutaneously injected with cancer spheroids at the same time and place (Supplementary Fig. 8). After confirming the anti-tumor effect of TIL-iPS-T in NSG mice with the Winn assay, we validated the therapeutic potential of TIL-iPS-T in PDSX mice. TI-CTL and TIL-iPS-T were transferred to PDSX mice in a multiclonal state containing all obtained tumor-specific clones (C-T-10 Vβ1, Vβ2, Vβ4, and Vβ9). TIL-iPS-T have the advantage of unlimited number, thus, repetitive TIL-iPS-T transfusion was allowed, but TI-CTL were transfused only once (Supplementary Fig. 9a). Blood samplings were done just before every cell transfusion to monitor the trough values of human CD45$^+$ cells, an indicator of persistency (Supplementary Fig. 9b). TIL-iPS-T did not exhibit a significant impact on tumor volume, but the luminescence levels of the inoculated tumors were significantly lower in the TIL-iPS-T group than the PBS group (Supplementary Fig. 9c, d, e). There was no difference in the luminescence levels between the TIL-iPS-T and TI-CTL groups. Ultimately, TIL-iPS-T did not prolong overall survival (Supplementary Fig. 9f).

Following those observations, we modified our protocol to improve the persistency of the transferred cells and evaluated the in vivo dynamics of TIL-iPS-T (Fig. 7a). The modified protocol and supplemented cytokines significantly improved persistency, especially of TIL-iPS-T (Fig. 7b). Bioluminescence tracing of the transferred cells revealed that TIL-iPS-T was able to accumulate at the tumor and remain there longer than TI-CTL (Fig. 7c). We confirmed that both TI-CTL and TIL-iPS-T infiltrated into tumor parenchyma by histological examination (Fig. 7d).

## Discussion

TIL were previously reprogrammed into iPS cells (TIL-iPSC)[60]. However, here we report for the first time the differentiation of T cells from TIL-iPSC. Moreover, we demonstrated stable multiclonal tumor-specific TIL regeneration for human colorectal cancers irrespective of the MMR profile.

The majority of studies have employed predictive surface markers[44,51,53,54] or peptide pulsing[12,21,23,26,51,65,66] for the tumor-specific TIL selection. On the other hand, we co-cultured the TIL with autologous cancer spheroids[67], because spheroids culture allowed us to access autologous cancer cells easily[40]. Furthermore, a Vβ-based cloning strategy was used for the multiclonal regeneration, because many rounds of interclonal competition, which

results in the diversity loss of tumor-specific clones, are inevitable during the long process of TIL regeneration. Our proposed method for predicting tumor-specific TI-CTL on a per-Vβ basis is simple, as it only blocks the TCR-HLA class I axis during co-culture but maintains the multiclonality of the final products. We confirmed the usefulness of our tumor-specific TI-CTL cloning method in multiclonal TIL regeneration, leading us to expect that this method will be helpful for ordinal TIL-ACT as well.

Several groups including ours have reported T-cell regeneration from iPSC by applying OF protocols[37,38,57–59], in which T-cell progenitors easily pass the DP stage. Because the quality of the regenerated T cells seems to be correlated with whether they experience the DP stage or not, OF protocols tend to produce T cells of high quality expressing CD8αβ heterodimers. On the other hand, FF protocols, which are required for clinical application, often induce T-cell progenitors to go straight to the SP cell stage from the CD4$^-$CD8$^-$ population, and the regenerated T cells frequently show features of innate lymphoid cells with CD8αα homodimers. However, we could obtain T cells of high quality from some TIL-iPSC clones which passed the DP stage. Of note, no additional TCR rearrangement was found in TIL-iPS-T induced from FF protocols.

Although we achieved stable multiclonal tumoricidal TIL regeneration, the reasons for the absence of additional TCR rearrangements in TIL-iPS-T are unknown. *RAG* expression during the DP stage did not differ between OF and FF protocols. The expression of other elements like E2A, which is involved in TCR rearrangement[68], may differ between the protocols. One difference we did find between protocols was that OF protocols tended to produce T-cell progenitors that remained in the DP stage until the maturation process began, whereas the same progenitors in FF protocols spontaneously formed a SP population. We assume that FF protocols possess some factors that serve for the production of SP cells, thus, DP cells change into SP cells before an additional TCR rearrangement occurs. Further investigation is required.

From the profiling analysis of TIL-iPS-T, we found that TIL-iPS-T exhibited some memory phenotype-related markers and characteristics unlike the parent TI-CTL. Moreover, we found that TIL-iPS-T retained not only intrinsic T-cell function and tumor specificity, but also acquired improved T cell- and persistency-related functions. Regarding the killing function, we confirmed that TIL-iPS-T utilized TCR-independent cytotoxicity in addition to the ordinal TCR-dependent mechanism. Because TIL-iPS-T did not show obvious killing activity against non-cancer spheroids or adverse effects on mice, we concluded that the TCR-independent cytotoxicity is not necessarily unfavorable. Overall, our data suggest that TIL-iPS-T could be a promising cell resource for TIL-ACT.

There are several limitations to this study. First, the clonality of tumor-specific TIL-iPS-T is not equivalent to that of ordinal TIL applied in clinical practice. Even though we achieved multiclonal TIL regeneration, TIL-iPS-T could have a narrower repertoire, because our tumor-specific TI-CTL cloning was based on only the TCR Vβ pattern. To reproduce the heterogeneity of ordinal TIL-

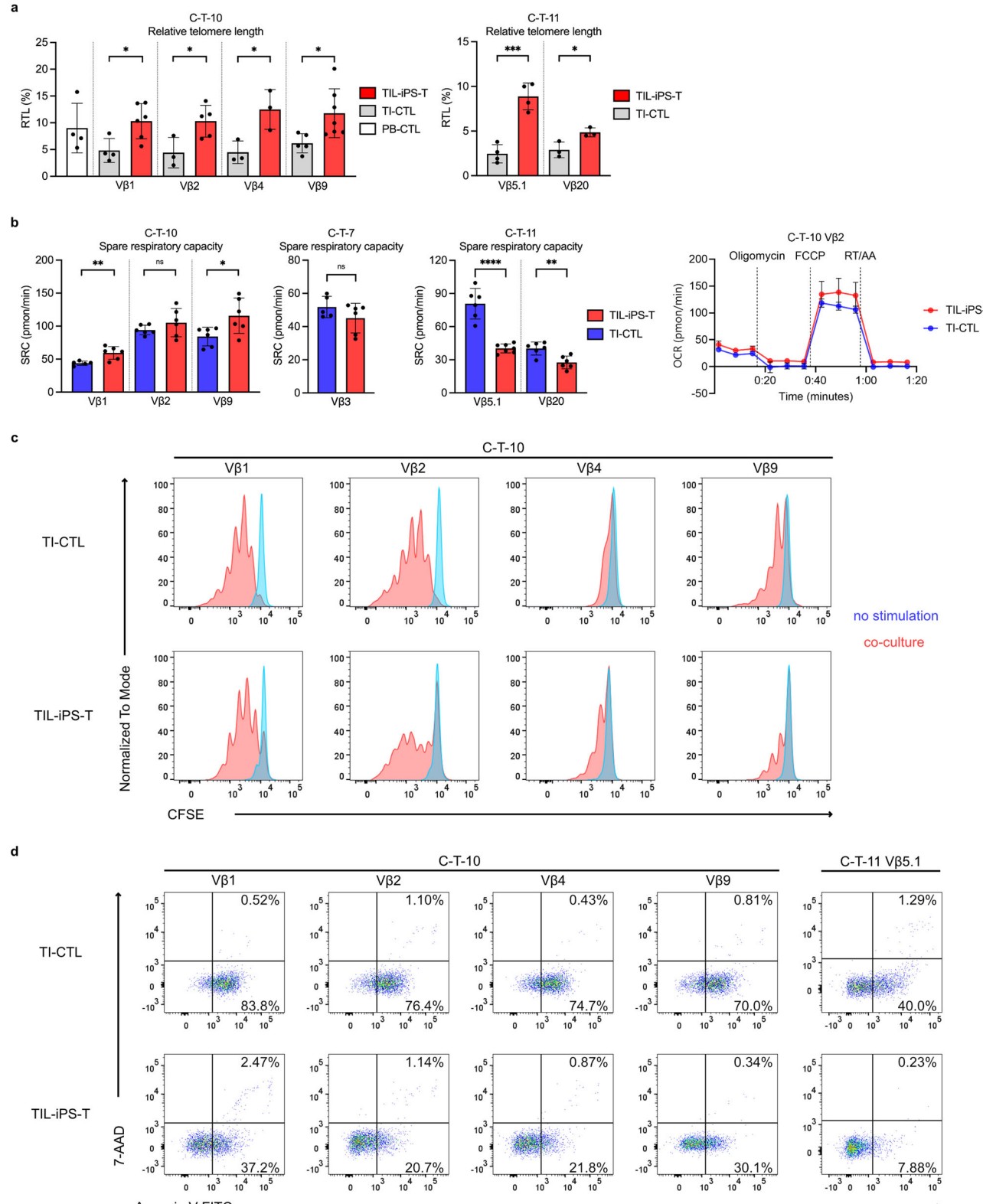

ACT in TIL-iPS-T, tumor-specific TI-CTL selection technologies that take all TCR sequence information into account, preferably from non-biased populations, are warranted. Second is the versatility of the TIL regeneration technology. Though we could regenerate functional T cells from almost all selected TI-CTL, this study deals only with colorectal cancer. The applicability of TIL-iPS-T to other malignancies should be assessed. Finally, TIL-iPS-T did not show obvious therapeutic effects in PDSX mice even though positive results were found in their dynamics. The PDSX model is regarded as an alternative to the patient-derived xenograft (PDX) model, but its utility has been reported only for sensitivity prediction to chemotherapy[69]. Because we do not have any positive control using the PDSX model for cell-based immunotherapy, it is difficult to conclude whether this model is

**Fig. 6 Persistency-related profiles of regenerated TIL-iPS-T. a** Bar graphs depict the relative telomere length (RTL) of TI-CTL and TIL-iPS-T. *$P < 0.05$, ***$P < 0.001$, two-tailed unpaired $t$ test. Dots represent data from individual experiments. Means are shown; error bars represent SD. **b** Representative mitochondria function analysis from two independent experiments are shown. Left: Bar graphs depict the SRC of TI-CTL and TIL-iPS-T. *$P < 0.05$, **$P < 0.01$, ****$P < 0.0001$, NS, not significant, two-tailed unpaired $t$ test. Dots represent individual values. Means are shown; error bars represent SD. Right: Oxygen consumption rate (OCR) trace data for C-T-10 Vβ2 are shown. Means are shown with dots and connected by lines; error bars represent SD. **c** Cell division capacity of TI-CTL and TIL-iPS-T were assessed by the CFSE-dilution assay. CFSE-labeled cells were co-cultured with cancer spheroids for 7 days. Representative data of two independent experiments are shown. Data were gated on CD45+ live cells. **d** Flow cytometry data about apoptosis sensitivity are indicated. Both TI-CTL and TIL-iPS-T were co-cultured with autologous cancer spheroids for 48 h. Representative data of three independent experiments are shown. Data were gated on CD45+ cells.

suitable or not at present. Further investigation and refinement of the PDSX model are needed.

In addition, efforts to enhance the therapeutic effect of TIL-iPS-T are needed. One possible approach is to transduce exogenous cytokines, such as membrane-bound IL-15 gene[62] and IL-12 mRNA[70]. Combination therapies with immune checkpoint inhibitors, irradiation, and adjuvant compounds are other options. In terms of the administration route, intratumor injections could be a solution as well, because it can induce not only local responses but also systemic ones[70].

To conclude, we demonstrated a proof of concept regarding multiclonal tumoricidal TIL-iPS-T regeneration. Our findings imply that iPSC technology has great potential for pushing TIL-ACT to the next stage.

## Methods

**Patient characteristics and materials.** Colorectal cancer patients ($n = 16$) who underwent primary tumor resections between December 2017 and November 2018 were enrolled in this study. Their clinical characteristics are detailed in Supplementary Table 1. This study was approved by the institutional review board of the Graduate School of Medicine, Kyoto University (Approval number: G590), and conducted in compliance with the Declaration of Helsinki. Both oral and written informed consent were obtained from all patients prior to the enrollment. All patient-derived materials are available only for this study. Clinical and pathological information about the patients was acquired from clinical records at Kyoto University Hospital. Four to eight pieces of 5-mm³ tumor fragments were collected from primary tumors immediately after surgical resection, and whole blood was collected on the same day. Plasma and the PBMC fraction were separated from whole blood using Leucosep™ (Greiner). Serum was purified from the separated plasma after heat inactivation and the removal of fibrin clots.

**Cell lines.** 1301, a subline of the Epstein–Barr virus genome negative T-cell leukemia line, was commercially obtained from ECACC and used for measuring RTL. TKT3V1–7 and GPC3 16-1 are iPS cell lines previously established (Nishimura et al.[38]; Minagawa et al.[59]).

**TIL isolation.** Obtained tumor fragments were washed in TIL buffer consisting of a 50/50 mix of RPMI1640 (Sigma-Aldrich)/D-PBS (Nacalai) supplemented with 1 mM EDTA (Nacalai), 0.5% human serum albumin (CSL Behring), 1% penicillin–streptomycin–amphotericin B suspension (Wako) or 1% antibiotic–antimycotic mixed stock solution (Nacalai), and 50 ng/ml gentamicin sulfate (Nacalai). Tumor fragments were minced with scissors, and the slurry was filtered with a 100-μm cell strainer (Greiner). A part of the cell suspensions passing through the cell strainer was characterized for flow cytometry. The residual cell suspensions were used for the TIL expansion with cytokine cocktails.

**TIL expansion with cytokine cocktails.** Cell suspensions after TIL isolation were cultured in TIL expansion medium with cytokines for 14 days. The TIL expansion medium consisted of MEM alpha (Invitrogen), 15% purified autologous serum or pooled heat-inactivated human AB serum (COSMO BIO), 2 mM GlutaMAX (Thermo Fisher Scientific), 1% insulin–transferrin–selenium (Thermo Fisher Scientific), 50 ng/ml L-ascorbic acid 2-phosphate (Nacalai), 0.5% 45 w/v% D (+)-glucose (Wako), 1% penicillin–streptomycin–amphotericin B suspension or 1% antibiotic–antimycotic mixed stock solution, and 50 ng/ml gentamicin sulfate. TIL were expanded in three cytokine conditions: IL-2 alone, IL-2 + IL-7 + IL-15 + IL-21, and IL-2 + IL-7 + IL-15 + IL-21 + IL-12 + IL-18. The final concentrations of the cytokines were 6000 U/ml IL-2 (Peprotech), 5 ng/ml IL-7 (Peprotech), 5 ng/ml IL-15 (Peprotech), 20 ng/ml IL-21 (Peprotech), 50 ng/ml IL-12 (Merck), and 50 ng/ml IL-18 (MBL). The cultures were split or scaled up to maintain less than 80% confluence or half of the culture medium was replaced every 2 to 3 days. Cells were counted in trypan blue and analyzed by flow cytometry after 14 days of expansion.

**Autologous cancer spheroids establishment and culture.** Residual tumor fragments that could not pass through the 100-μm cell strainer were collected and enzymatically digested using collagenase solution consisting of spheroids wash medium supplemented with 2 mg/ml collagenase type I (Thermo Fisher Scientific) solution. Spheroids wash medium consisted of DMEM/F-12 (Sigma-Aldrich), 10% fetal bovine serum (Sigma-Aldrich), 2 mM GlutaMAX, 1% penicillin–streptomycin–amphotericin B suspension or 1% antibiotic–antimycotic mixed stock solution, and 50 ng/ml gentamicin sulfate. Digested cell pellets were collected after filtering with the 100-μm cell strainer and removing collagenase solution. Cell pellets were embedded in Matrigel (Corning) and placed in the center of the wells. After the polymerization of Matrigel for 10 min at 37 °C, the embedded cells were overlaid with spheroids culture medium supplemented with 10 μM Y-27632 (Wako) and 1 μM SB431542 (Cayman Chemical) for the first 2 days. Spheroids culture medium consisted of advanced DMEM/F-12 (Thermo Fisher Scientific), 5% fetal bovine serum, 2 mM GlutaMAX, 1% penicillin–streptomycin–amphotericin B suspension or 1% antibiotic–antimycotic mixed stock solution, 50 ng/ml gentamicin sulfate, 50 ng/ml human EGF (Peprotech) and 100 ng/ml bFGF (Wako). The culture medium was replaced every other day with spheroids culture medium. Cancer spheroids were trypsinized and embedded in Matrigel again for passaging.

**Autologous non-cancer spheroids culture.** Non-cancer spheroids were embedded in Matrigel and placed in the center of the wells. After the polymerization of Matrigel for 10 min at 37 °C, the embedded cells were overlaid with IntestiCult Organoid Growth Medium (STEMCELL Technologies) supplemented with 10 μM Y-27632 and 1 μM SB431542. The culture medium was replaced every other day with IntestiCult Organoid Growth Medium. Non-cancer spheroids were trypsinized and embedded in Matrigel again for passaging.

**Anti-human antibodies for flow cytometry.** The following anti-human antibodies used for the flow cytometry were purchased.

*From BioLegend.* CD3-BV510 (clone: UCHT1, 300448), CD3-APC (clone: UCHT1, 300412), CD3-APC-Cy7 (clone: UCHT1, 300426), CD4-BV421 (clone: OKT4, 317434), CD7-FITC (clone: CD7-6B7, 343104), CD8-PerCP-Cy5.5 (clone: SK1, 344710), CD14-PE-Cy7 (clone: HCD14, 325618), CD16-FITC (clone: 3G8, 302006), CD27-APC (clone: O323, 302810), CD28-BV421 (clone: CD28.2, 302930), CD34-PB (clone: 581, 343512), CD45-BV421 (clone: HI30, 304031), CD45-BV510 (clone: HI30, 304036), CD45RA-BV510 (clone: HI100, 304142), CD56-APC-Cy7 (clone: HCD56, 318332), CD62L-PE-Cy7 (clone: DREG-56, 304822), CD95-PE-Cy7 (clone: DX2, 305622), CD107a-PE (clone: H4A3, 320608), 4-1BB (CD137)-PE (clone: 4B4-1, 309804), 4-1BB (CD137)-APC (clone: 4B4-1, 309810), CD158-APC (clone: HP-MA4, 339509), CD158a/h-FITC (clone: HP-MA4, 339503), CD158b-APC (clone: DX27, 312715), CD158e1-APC (clone: DX9, 312715), CD161-PE-Cy7 (clone: HP-3G10, 339918), CCR7 (CD197)-APC (clone: G043H7, 353214), LAG-3 (CD223)-FITC (clone: 11C3C65, 369308), CD226 (DNAM-1)-BV421 (clone: 11A8, 338332), PD-1 (CD279)-BV421 (clone: EH12.2H7, 329920), CD314 (NKG2D)-PE-Cy7 (clone: 1D11, 320811), CD335-FITC (clone: 29A1.4, 331921), CD336-APC (clone: P44-8, 325110), CD337-APC (clone: P30-15, 325209), Tim-3 (CD366)-PE-Cy7 (clone: F38-2E2, 345014), IL-2-PE (clone: MQ1-17H12, 500307), INF-γ-APC-Cy7 (clone: B27, 506524), TCF1 (TCF7)-AF647 (clone: 7F11A10, 655203), and TCRαβ-APC (clone: IP26, 306718).

*From BD Biosciences.* CD4-APC-H7 (clone: RPA-T4, 560158), CD45-APC (clone: HI30, 555485), CD107a-APC (clone: H4A3, 560664), CD144-FITC (clone: 55-7H1, 560411), CD159a (NKG2A)-BV421 (clone: 131411, 747924), CD235a-APC (clone: GA-R2 (HIR2), 551336), and IL-2-APC (clone: MQ1-17H12, 561054).

*From eBioscience.* CD5-PE-Cy7 (clone: UCHT2, 25-0059-42) and CD43-PE (clone: eBio84-3C1, 12-0439-42).

*From Beckman Coulter.* CD8β-PE (clone: 2ST8.5H7, IM2217U).

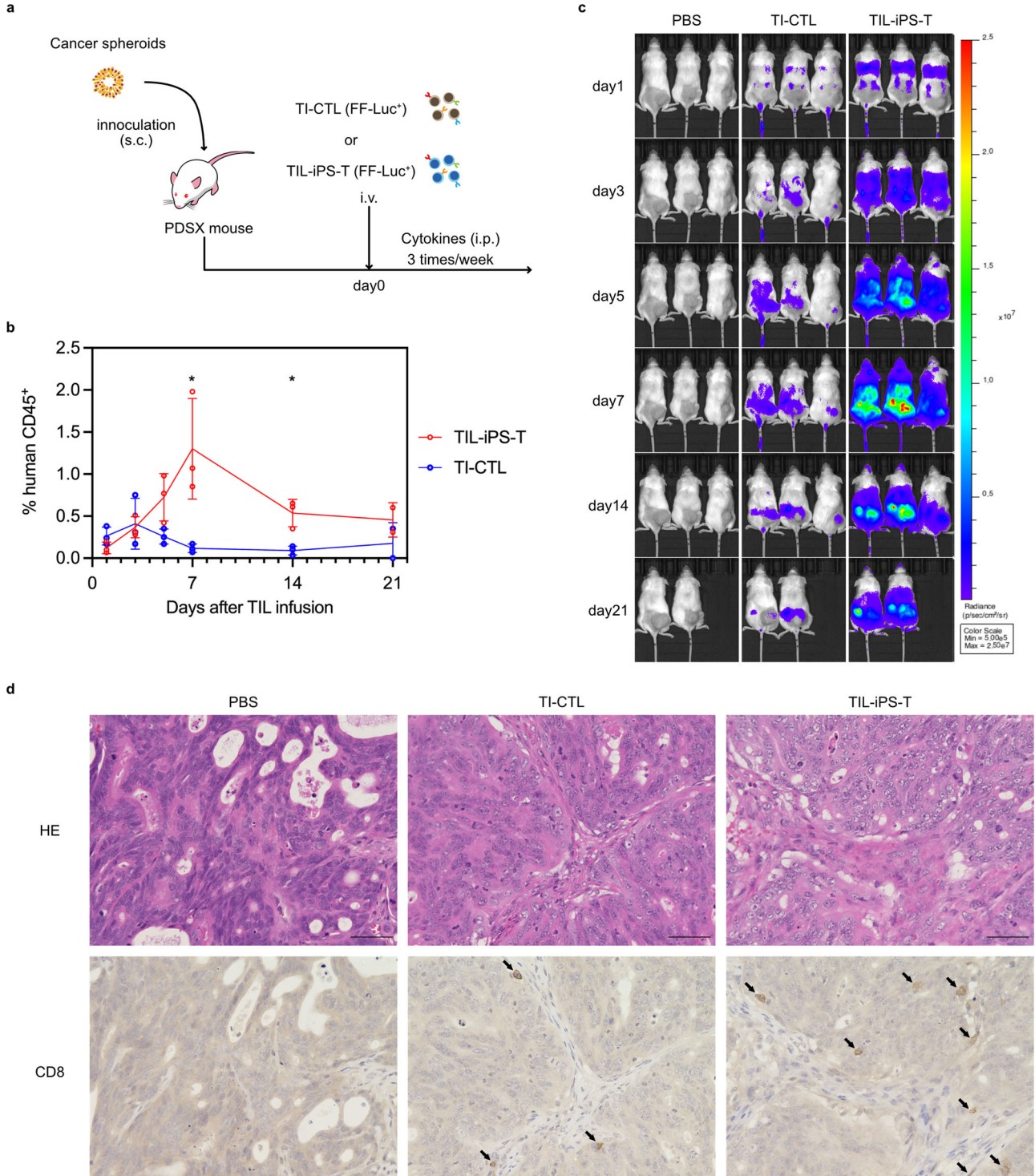

**Fig. 7 In vivo dynamics of TIL-iPS-T. a** A schema of the experiment design. Non-labeled cancer spheroids were used for the PDSX establishment. In total, $7.5 \times 10^6$ firefly luciferase-transduced (FF-Luc+) TI-CTL or TIL-iPS-T were intravenously injected into PDSX mice. Cytokines (IL-2 200 U/mouse, IL-7 50 ng/mouse, IL-15 50 ng/mouse) were intraperitoneally injected three times a week. **b** The percentages of human CD45+ cells within the mouse PBMC fraction are indicated. Dots represent individual values. Means are connected by lines; error bars represent SD. $n = 3$ per point except day 21 ($n = 2$). *$P < 0.05$, two-tailed unpaired $t$ test. **c** Bioluminescence images by an in vivo imaging system (IVIS) are shown. Bioluminescence signals were captured from FF-Luc+ TI-CTL or TIL-iPS-T. **d** Histological images of an isolated tumor on day 14 are shown. Upper: hematoxylin eosin (HE) staining. Lower: immunostaining of human CD8. Arrows indicate TI-CTL or TIL-iPS-T stained with CD8 antibody. Scale bars indicate 50 µm.

**Flow cytometry analysis.** For surface staining, cells were washed in PBS supplemented with 2% fetal bovine serum and stained with the appropriate antibodies for 30 min at 4 °C in the dark. The dilutions of antibodies are provided in Supplementary Table 3. After the surface staining, the cells were washed, and 1 µg/ml propidium iodide (Sigma-Aldrich) was added to exclude dead cells. To access the TCR Vβ repertoire, the Beta Mark TCR Vβ Repertoire Kit (IM3497, Beckman Coulter) was used. For intracellular staining, reagents (Fixation Buffer, Cell Staining Buffer, and Permeabilization Buffer) and True-Nuclear Transcription Factor Buffer set purchased from BioLegend were used. Data were acquired on an LSRFortessa (BD) or FACSAria flow cytometer (BD) and analyzed with FlowJo software (Tree Star).

**Reactivity assays against spheroids.** After pre-treatment with 1000 U/ml INF-γ (Peprotech) for 48 h, spheroids were dissociated into single cells by trypsinizing. To block HLA class I, dissociated cells were pre-incubated for 30 min with 50 µg/ml HLA class I blocking antibody (clone: w6/32, BioLegend, 311428); the antibody was continuously added during the co-culture. Spheroids and TI-CTL or TIL-iPS-T were co-cultured at a ratio of 1:3 in 96-well U-bottom plates for 5 h for the CD107a assay and for 24 h for the 4-1BB assay. OP9 medium supplemented with 100 U/ml IL-2, 5 ng/ml IL-7, and 5 ng/ml IL-15 was used for the co-culture. OP9 medium consisted of MEM alpha, 15% fetal bovine serum, 2 mM L-glutamine, 100 U/ml penicillin and 100 ng/ml streptomycin (Sigma-Aldrich). CD107a antibodies and 2 µM monensin (BioLegend) were added from the beginning of the co-culture for the CD107a assay. After finishing the co-culture, CD45 was stained in both assays, and 4-1BB was stained in the 4-1BB assay for the flow cytometry analysis.

**Killing assays against spheroids.** Spheroids were dissociated with trypsin before the killing assays. The following assays were used to measure the killing functions of TI-CTL and TIL-iPS-T.

*$^{51}$Cr release assay.* Dissociated spheroids were pre-incubated with 925 kBq/ml of $^{51}$Cr solution (PerkinElmer) for 1 h. After washing out residual $^{51}$Cr solution, the target cells were co-cultured with effector cells for 5 h in 96-well U-bottom plates at the indicated effector/target (E/T) ratios. The target cell number was fixed at 5000 cells/well. TIL culture medium supplemented with 100 U/ml IL-2, 5 ng/ml IL-7, and 5 ng/ml IL-15 was used for the co-culture. The released $^{51}$Cr was measured by MicroBeta2 (PerkinElmer).

*Luminescence-based killing assay.* Firefly luciferase was transduced into spheroids with lentiviruses. The FF-Luc$^+$ dissociated spheroids were co-cultured with effector cells in a 96-well CELLSTAR microplate (Greiner) at the indicated E/T ratios. The co-culture time was 8 h in C-T-7 and C-T-10, and 12 h in C-T-11. The target cell number was fixed at 5000 cells/well. Phenol red-free D-MEM (Wako) supplemented with 10% fetal bovine serum, 2 mM L-glutamine, 100 U/ml penicillin, 100 ng/ml streptomycin, 100 U/ml IL-2, 5 ng/ml IL-7, and 5 ng/ml IL-15 was used for the co-culture. In total, 150 µg/ml VivoGlo Luciferin (Promega) was added to each well after the co-culture, and luminescence levels were measured using Centro LB960 (BERTHOLD).

**iPSC generation from TI-CTL.** A few days prior to the TI-CTL stimulation, the culture medium was changed to reprogramming medium consisting of MEM alpha supplemented with 15% fetal bovine serum, 2 mM L-glutamine, 100 U/ml penicillin, 100 ng/ml streptomycin, 50 µg/ml L-ascorbic acid 2-phosphate, 50 U/ml IL-2, and 10 ng/ml IL-15. After the pre-culture, TI-CTL were stimulated with plate-bound CD3 Ab (clone: OKT3, BioLegend) and 1 µg/ml soluble CD28 Ab (clone: CD28.2, BioLegend). After 2 days of stimulation, $1 \times 10^4$ TI-CTL were transfected with Sendai virus vectors containing reprogramming factors (CytoTune®-iPSC2.0, ID Pharma) and SV40LT (ID Pharma) at a multiplicity of infection (MOI) of 20. After 24 h of transfection, the culture medium was totally changed, and transfected TI-CTL were seeded on Easy iMatrix-511 silk (Nacalai)-coated plates. After another 24-h incubation under normoxia (21% $O_2$), TI-CTL were transferred to hypoxic culture condition (5% $O_2$). AK02 (Ajinomoto) supplemented with 1 mM valproic acid (Wako) was added every other day during the hypoxic culture condition. On day 8 after the transfection, the culture condition was returned to normoxia. The culture medium was replaced with AK02 with valproic acid every other day until day 14. On day 14, generated colonies were counted and picked up.

**T-cell differentiation.** The T-cell differentiation steps are composed of two phases. The first phase is HPC induction and the second phase is T-cell lineage induction. Both OF and FF protocols are available. After the two phases, the induced T cells undergo the maturation process.

*On-feeder protocols.* Clumps of T-iPSC were transferred onto C3H10T1/2 feeder cells and cultured in Sac medium supplemented with 20 ng/ml BMP-4 (R&D) under hypoxia (5% $O_2$) for 6 days. Sac medium consists of IMDM (Sigma-Aldrich) supplemented with 15% fetal bovine serum, 2 mM L-glutamine, 100 U/ml penicillin, 100 ng/ml streptomycin, 1% insulin–transferrin–selenium, 450 mM mono-thioglycerol (Nacalai), 50 µg/ml L-ascorbic acid 2-phosphate, and 50 ng/ml VEGF (Wako). The culture condition was returned to normoxia after 6 days hypoxic culture and changed to Sac medium supplemented with 30 ng/ml SCF (R&D) and 10 ng/ml Flt3L (Peprotech). On day 14, HPC were collected and transferred onto OP9DL1 cells in OP9 medium supplemented with 1% insulin–transferrin–selenium, 50 µg/ml L-ascorbic acid 2-phosphate, 1 ng/ml IL-7, and 10 ng/ml Flt3L. After 3 weeks culture on OP9DL1, DP cells underwent the maturation process.

*Feeder-free protocols.* iPSC were dissociated with TrypLE select Enzyme (Thermo Fisher Scientific) and cultured in AK02 supplemented with 10 µM Y-27632 and 3 µM CHIR99021 (Tocris) under hypoxia (5% $O_2$) using ultralow attachment microplates (Corning). On day 1, the culture medium was changed to EB medium supplemented with 40 ng/ml BMP-4, 50 ng/ml bFGF, and 50 ng/ml VEGF. EB medium consists of StemPro-34 (Invitrogen), 2 mM GlutaMAX, 100 U/ml penicillin, 100 ng/ml streptomycin, 1% insulin–transferrin–selenium, 400 µM mono-thioglycerol, and 50 µg/ml L-ascorbic acid 2-phosphate. On day 2, 6 µM SB431542 (Cayman) was added to the culture medium. On day 4, the culture medium was changed with EB medium supplemented with 50 ng/ml bFGF, 50 ng/ml VEGF, and 50 ng/ml SCF. On day 6, the culture condition was returned to normoxia, and the medium was changed to EB medium supplemented with 50 ng/ml bFGF, 50 ng/ml VEGF, 50 ng/ml SCF, 10 ng/ml Flt3L, and 30 ng/ml TPO (Peprotech). On day 14, HPC (CD14$^-$CD235a$^-$CD34$^+$CD43$^+$) were collected and transferred onto FcDLL4-coated plates filled with OP9 medium supplemented with 1% insulin–transferrin–selenium, 50 µg/ml L-ascorbic acid 2-phosphate, 50 ng/ml IL-7, 55 µM 2-mercaptoethanol (Invitrogen), 50 ng/ml SCF, 50 ng/ml Flt3L, 100 ng/ml TPO, 15 µM SB203580 (TOCRIS), and 30 nM SDF-1α (Peprotech). FcDLL4-coated plates were prepared with 5 µg/ml FcDLL4 (Cosmobio) dissolved in 5 µg/ml Ret-roNectine (TAKARA Bio) at 4 °C overnight. The medium was changed every other day, and FcDLL4-coated plates were replaced every week. After 3 weeks of culture on FcDLL4, T-cell progenitors underwent the maturation process.

*Maturation process.* After the T-cell induction, the cells were stimulated in OP9 medium supplemented with 1% insulin–transferrin–selenium, 50 µg/ml L-ascorbic acid 2-phosphate, 10 ng/ml IL-7, 10 ng/ml Flt3L, and 1 µg/ml soluble CD3 Ab (clone: OKT3, BioLegend). After 2 days of stimulation, the cells were transferred onto a RetroNectine-coated plate and cultured in OP9 medium supplemented with 1% insulin–transferrin–selenium, 50 µg/ml L-ascorbic acid 2-phosphate, 10 ng/ml IL-7, 10 ng/ml Flt3L, and 10 ng/ml IL-21. The generated T cells with CD8αβ heterodimers were selectively expanded with the PHA/PBMC method.

**PHA/PBMC expansion method.** T cells were co-cultured and irradiated (40 Gy) on allogeneic PBMC serving as feeder cells at a ratio of 1:10 in OP9 medium supplemented with 50 µg/ml L-ascorbic acid 2-phosphate, 5 ng/ml IL-7, 5 ng/ml IL-15, and 2 µg/ml PHA (WAKO). Half of the medium was changed every other day without PHA.

**NGS repertoire analysis of TIL-iPS-T.** Live TIL-iPS-T were sorted, and the total RNA was isolated from them. An NGS-based TCRα and TCRβ repertoire analysis was performed by Repertoire Genesis Incorporation (Osaka, Japan). Out-of-frame sequences were excluded from the analysis.

**Quantitative PCR (qPCR).** The total RNA was isolated using the RNeasy Micro Kit (QIAGEN), and cDNA was synthesized using the High-Capacity cDNA Reverse Transcription Kit (Applied Biosystems) according to the manufacturer's instructions. qPCR was performed on StepOnePlus (Applied Biosystems). Data were analyzed by the $2^{-\Delta\Delta Ct}$ method.

*RAG expression in T-cell progenitors.* DP cells from both OF and FF protocols were sorted at different time points of the T-cell induction. cDNA was synthesized from the DP cells. qPCR was performed using TB Green® Premix Ex Taq (TaKaRa) according to the manufacturer's protocol. The primer sequences are listed in Supplementary Table 4. The primers were custom designed and synthesized by FASMAC.

*TCF-1 (TCF-7) expression in TIL-iPS-T.* CD45$^+$CD4$^-$CD8α$^+$CD8β$^+$ live cell populations of both TI-CTL and TIL-iPS-T 21 days after culturing on FcDLL4 were sorted for cDNA synthesis. TaqMan$^{TM}$ Fast Universal PCR Master Mix was used for qPCR according to the manufacturer's protocol. TaqMan probes of ACTB (4331182) and TCF7 (4331182) were purchased from Applied Biosystems.

**Assays for cytokine production.** T cells were stimulated with 50 ng/ml PMA (Wako) plus 1 µg/ml ION (Wako) or co-cultured with cancer spheroids. 100,000 T cells and cancer spheroids each were plated on 96-well U-bottom plates. OP9 medium was used for the assays in the presence of 5 ng/ml IL-7 and 5 ng/ml IL-15.

*Intracellular staining.* In all, 2 µM monensin was supplemented in the medium before stimulation. After 5 h of stimulation, the cells were collected and stained for extracellular molecules. After the extracellular staining, the cells were fixed and permeabilized overnight. On the following day, intracellular cytokines were stained and analyzed by flow cytometry. All products related to the intracellular staining were purchased from BioLegend (Fixation Buffer, Cell Staining Buffer and Intra-cellular Staining Permeabilization Wash Buffer).

*Cytometric beads array (CBA).* The culture supernatant was collected 48 h after the stimulation. CBA was performed with the Human Th1/Th2/Th17 Kit (BD) according to the manufacturer's instruction. The acquired data were analyzed using FCAP Array Software v3.0 (BD).

**Measuring relative telomere length (RTL)**. RTL was measured using the Telomere PNA Kit/FITC for Flow Cytometry (DAKO) according to the manufacturer's protocol. 1301 was used as the control cell line to calculate the RTL of TI-CTL and TIL-iPS-T.

**Metabolism assay**. Measurement of the OCR was performed on an XF96 Extracellular Flux Analyzer (Seahorse Bioscience). In total, 200,000 cells were plated in each well filled with non-buffered RPMI1640 (Sigma-Aldrich, R6504) containing 1 mM sodium pyruvate (Nacalai), 5 ng/ml IL-7, and 5 ng/ml IL-15. After preincubation under a non-$CO_2$ condition for 30 min at 37 °C, the cells received sequential injections of 1 μM oligomycin, 2.2 μM fluorocarbonyl cyanide phenylhydrazone (FCCP), and 500 nM rotenone/antimycin A. The acquired data were analyzed using Wave Desktop 2.4 (Agilent Technologies).

## Evaluation of mitochondria biomass
Cells were stained with 100 nM MitoTracker™ Deep Red FM (Thermo Fisher Scientific) according to the manufacturer's protocol. Data were acquired by flow cytometry.

**Apoptosis assay**. T cells were co-cultured with cancer spheroids for 48 h in OP9 medium supplemented with 5 ng/ml IL-7 and 5 ng/ml IL-15. In total, 100,000 T cells and cancer spheroids each were plated in 96-well U-bottom plates. An FITC Annexin V Apoptosis Detection Kit with 7-AAD (BioLegend, 640922) was used for the assay. The cells were stained according to the manufacturer's protocol.

**CFSE-dilution assay**. T cells were labeled with CFSE using the CFSE Cell Division Assay Kit (Cayman) according to the manufacturer's protocol. In total, 100,000 T cells and cancer spheroids each were co-cultured in 96-well U-bottom plates for 7 days. The cells were cultured in OP9 medium supplemented with 5 ng/ml IL-7 and 5 ng/ml IL-15. CFSE dilution was detected by flow cytometry.

**Time-lapse killing image of TIL-iPS-T**. TIL-iPS-T were co-cultured with GFP-transduced cancer spheroids at a ratio of 1:1 in phenol red-free D-MEM supplemented with 10% fetal bovine serum, 2 mM L-glutamine, 100 U/ml penicillin, 100 ng/ml streptomycin, 100 U/ml IL-2, 5 ng/ml IL-7, and 5 ng/ml IL-15. In all, 1 μg/ml propidium iodide was also added to stain dead cells. Image capturing was performed on BioStation IM-Q (NICON) for 24 h. The captured images were edited using EDIUS Neo 3.5 (Grass Valley).

**PDSX mice**. 5- or 6-week-old female NOD-SCID IL2Rγcnull (NSG) mice were purchased from Oriental Bio (Yokohama, Japan). The mice were housed under controlled conditions, humidity, and a light/dark cycle in a specific-pathogen-free facility. All animal experiments and procedures were permitted by the Kyoto University Institutional Review Board. In all, $1 \times 10^6$ cancer spheroids dissociated with trypsin were subcutaneously injected into NSG mice to establish passage 0 (P0) PDSX mice. The engrafted tumors were extracted, minced, and subcutaneously transplanted into new NSG mice to maintain PDSX mice.

**Evaluating anti-tumor efficacy of TIL-iPS-T in PDSX mice (Supplementary Fig. 9)**. FF-Luc+ cancer spheroids for C-T-10 MMR-D were used to establish PDSX mice. In all, $5 \times 10^6$ TI-CTL were transferred once. In total, $5 \times 10^6$ TIL-iPS-T were transferred the first five times and $1 \times 10^7$ cells the last three times. Thus, the total number of transferred TIL-iPS-T was $5.5 \times 10^7$ cells per mouse. All of the available clones (Vβ1, Vβ2, Vβ4, and Vβ9) were mixed at the same ratio and injected. Peripheral blood from PDSX mice was stained with anti-human CD45-APC and anti-mouse CD45-PE (BioLegend, 103106) antibodies for the flow

cytometry analysis. The euthanasia criteria consisted of either tumor necrosis, tumor ulceration, hypoactivity due to tumor growth, a tumor diameter greater than 20 mm and tumor volume greater than 2000 mm³.

**In vivo dynamics of TIL-iPS-T (Fig. 7)**. Non-labeled cancer spheroids were used to establish PDSX mice. In total, $7.5 \times 10^6$ FF-Luc+ TI-CTL or TIL-iPS-T from C-T-10 Vβ2 were transferred once. Cytokines (IL-2 200 U/mouse, IL-7 50 ng/mouse, IL-15 50 ng/mouse) were intraperitoneally injected three times a week. Peripheral blood from PDSX mice was stained with anti-human CD45-APC and anti-mouse CD45-PE antibodies for the flow cytometry analysis. On day 14, tumors from each group were excised for the histological evaluation staining with HE and human CD8 antibody (clone: C8/144B, DAKO, M7103). All mice were euthanized on day 21 because of the experiment end.

**In vivo bioluminescence imaging (IVIS)**. Bioluminescence image capturing was performed on an IVIS Spectrum (PerkinElmer) or IVIS Lumina (PerkinElmer) 10 min after the intraperitoneal injection of 1.5 mg/mouse VivoGlo Luciferin (Promega). The captured images were analyzed using Living Image software (PerkinElmer).

**Statistics and reproducibility**. Data analyses and representations were performed with Excel (Microsoft) and Prism 8 (GraphPad software). Statistical tests and the number of experiments and samples are specified in the figures or figure legends. Parametric data are presented as means ± SD, and other data as medians and interquartile ranges. All tests were two-tailed. $P$ values <0.05 were considered to be statistically significant.

**Reporting summary**. Further information on research design is available in the Nature Research Reporting Summary linked to this article.

## Data availability
The datasets generated and analyzed during this study are available from Supplementary Data 1–3 or from the corresponding author upon reasonable request.

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

## Acknowledgements
We thank Shinya Yamanaka for providing critical advice; Takehito Yamamoto for providing the spheroids; Kanae Mitsunaga for technical assistance with the flow cytometry analysis; Shunsuke Kihara for technical assistance with the time-lapse imaging; Yoriko Indo, Daisuke Seki and Akito Tanaka for supporting the animal experiments; Norikazu Saiki, Kohei Ohara, Eri Imai, Kaede Makino, Sanae Kamibayashi, Munehiro Yoshida, Kengo Nakagoshi, Shuichi Kitayama, Tadayo Miyasaka, Sayaka Okamoto, Ayako Kumagai, Yoshihiro Iwamoto, Yoshitaka Ishiguro, Masahiro Tanaka, Tatsuki Ueda, Masazumi Waseda, Akihiro Ishikawa, Bo Wang, Sara Shiina, Reiko Saikawa, Yuta Mishima, Hitomi Takakubo, Yukie Seto, and Katsura Noda for technical assistance; and Peter Karagiannis for editing the manuscript. This study was supported in part by a Grant-in-Aid for Scientific Research (KAKENHI) and collaborative research grant of Thyas Co., Ltd.

## Author contributions
T. Ito and S.K. designed the concept and study. T. Ito performed the experiments and acquired the data. Y.Y. cooperated with the iPSC establishment from TIL. T. Ito and S.K. analyzed and interpreted the data. Y.K., S.I., and A.M. provided protocols and advice for the experiments. T. Ishii cooperated with the in vivo experiments. H.M. and M.M.T. provided the spheroids and instructed on the spheroids culture methods. K.K., K.O., and Y.S. provided the human colorectal cancer specimens. S.K. supervised the study. T. Ito and S.K. wrote the manuscript.

## Competing interests
S.K. is a founder, shareholder, and chief scientific officer at Thyas Co., Ltd. and received research funding from Thyas Co., Ltd., Takeda Pharmaceutical Co., Ltd., Kirin Co., Ltd., TERUMO Co., Ltd., Astellas Pharma Co., Ltd. and Tosoh Co. Ltd. The remaining authors declare no competing interests.
