## [Peer Review File · Communications Biology]

Reviewers' comments:

Reviewer #1 (Remarks to the Author):

To the authors:

In this manuscript Ito et al. describe the potential of rejuvenating TILs by the process of reprogramming to iPSC cells. The authors have reprogrammed TILs derived from resected colorectal cancer samples and they show that the generated TiPSC can be differentiated towards TiPSC-TILs keeping the anti-tumor reactivity against autologous tumor spheroids. The rational and the potential translational benefit from the results of this study is clear, meaning that indeed autologous TIL therapy could be improved and it's applicability facilitated if there was unlimited availability of enriched and functionally proficient tumor specific TILs. The authors present a well thought process for the selection of tumor antigen specific TILs to be reprogrammed. However, many necessary TIL characteristics that would prove the feasibility of this concept are not demonstrated in this manuscript. Below are some of my major points of concern:

1. In order for the whole methodology to make sense then true multiclonal regeneration should be demonstrated. Multiclonality against various neoantigens is one of the preferred characteristics of TIL therapy. The authors should provide information on how many different and of good quality clones they can rejuvenate from one patient sample. What percentage of the initial polyclonal sample do they represent? The authors show maximum 4 clones from one patient sample. Would that be enough to replace ordinary TILs?
2. Most the TiPSC clones generated do not show improved cytotoxicity, cytokine production or persistence characteristics compared to the oligoclonal parental population. So there is no functional benefit of the rejuvenation process. The authors show cytokine production after unspecific stimulation with PMA/Ion in the main figure. However I think that the stimulation with autologous tumor should be more informative. In supplementary figure 5a none of the TiPSC-TILs shows IL-2 production after stimulation with tumor, similarly to the parental TILs. Thus, although the authors show immunophenotypic profiles of memory cells (CD62L+, CCR7+, CD28+ etc) the functional profile is far from that.
3. Even the TiPSC-TIL from MMR-P samples showed improved cytotoxicity against autologous tumor spheroids. The authors in the discussion attribute that to a NK-cell activity of the TiPSC-TIL. If that is the case then why the same NK-cell activity is not observed with the non-tumor specific clones? And for the supposedly tumor-specific clones do we know which part of the cytotoxicity is NK cell-like and which is mediated through the TCR? The HLA inhibition experiment used for the selection should be repeated for the TiPSC-TILs.
4. Since no benefit in persistence is observed compared to the parental TILs, one would expect that this at least should be overcome with the other benefit of reprogramming, the unlimited potential for dosing. However, even injection of 10 times more TiPSC-TILs did not result in a beneficial anti-tumor effect. It seems that the selected tumor model used to assess in vivo anti-tumor function is unsuccessful (since there is no positive control there) or that the generated TiPSC-TIL are actually defective in functionality. In any case there is no proof of a functional benefit through rejuvenation. In addition it is not clear which TiPSC clones were used in the in vivo experiment. Was it all 4 clones of C-T-10?

Reviewer #3 (Remarks to the Author):

While reprogramming of TIL to iPSC as a method to improve ACT has been reported, the current manuscript establishes novel methods of selecting tumor-specific CTL for reprogramming using autologous cancer spheroids. The authors then go on to further establish a protocol to regenerate stable multiclonal tumor-specific TIL (TIL-iPS-T). This is a well-written and logical study that presents a careful characterization of potentially important advances on ACT methodology.

Comments:

While this study provided a logical assessment of the novel differentiation processes, I have a couple questions regarding the heterogeneity of the populations that result:

1. HLA I blocking antibody only partially inhibits TI-CTL expansion leading the authors to speculate that some of the TI-CTLs (CD56+) exhibit NK-cell like functions. Further phenotyping of the heterogeneous cell population that results from autologous cancer spheroid TI-CTL enrichment may be important to define this novel means of enrichment.

2. The TIL-iPS-T seem to differ depending on tumor source. Does this protocol result in a different population of T cells depending on the source? In other words, does the degree of differentiation of the T cells population vary? Can you get a population of T cells that are more memory-like (or stem-like; TCF+) that may be able to proliferate more or have increased SRC. A phenotyping of the T cell differentiation states within the populations of TIL-iPS-T may be important to evaluate this.

Minor point:

- The authors evaluate the number of TI-CTL in a 5mm³ tumor block and conclude based on low number that no expansion is taking place in the tumor (page 7/line 15-16). While probably that most expansion takes place before tumor infiltration, I am not sure that tumor localized expansion can be ruled out solely based on numbers.

Point-by-point responses to the reviewer's comments:

We thank the reviewers for their helpful and constructive comments. We write our point-by-point responses to each comment below.

Reviewer #1 (Remarks to the Author):

To the authors:

In this manuscript Ito et al. describe the potential of rejuvenating TILs by the process of reprogramming to iPS cells. The authors have reprogrammed TILs derived from resected colorectal cancer samples and they show that the generated TiPSC can be differentiated towards TIPSC-TILs keeping the anti-tumor reactivity against autologous tumor spheroids. The rational and the potential translational benefit from the results of this study is clear, meaning that indeed autologous TIL therapy could be improved and it's applicability facilitated if there was unlimited availability of enriched and functionally proficient tumor specific TILs. The authors present a well thought process for the selection of tumor antigen specific TILs to be reprogrammed. However, many necessary TIL characteristics that would prove the feasibility of this concept are not demonstrated in this manuscript. Below are some of my major points of concern:

1. In order for the whole methodology to make sense then true multiclonal regeneration should be demonstrated. Multiclonality against various neoantigens is one of the preferred characteristics of TIL therapy. The authors should provide information on how many different and of good quality clones they can rejuvenate from one patient sample. What percentage of the initial polyclonal sample do they represent? The authors show maximum 4 clones from one patient sample. Would that be enough to replace ordinary TILs?

→ We would like to thank the reviewer for the comment. In this study, we intended to obtain tumor-specific TI-CTL from 3 patients. The number of acquired tumor-specific TI-CTL was 4 clones from C-T-10 and 2 clones each from C-T-7 and C-T-11. We could regenerate TIL that functioned as primary T cells from all selected clones except C-T-7 Vβ22. Information about the clones was added to Supplementary Table 2 and page 11/line 1-2 in the revised manuscript. The successful percentage of TIL regeneration from the

selected clones and tumor-specific clones was 90% (9/10) and 87.5% (7/8), respectively. Thus, the rate-limiting stage of the multiclonal TIL regeneration seems to be the tumor-specific TI-CTL cloning phase. Similar to the reviewer's concern, we also think that tumor-specific TIL-iPS-T are not equivalent to ordinal TIL with regards to their repertoire. Because our method utilizes only TCR V β information for the TI-CTL selection, it is likely that the diversity of regenerated TIL-iPS-T is narrower than that of ordinal TIL. The reports about TIL-ACT (Chatani et.al., *Clin. Cancer Res.* 2020 , Zacharakis et.al., *Nat. Med.* 2018 etc.) imply the necessity of targeting multiple antigens. Nevertheless, how many clones are desired for TIL-ACT is unknown. Thus, we cannot answer whether our multiclonal TIL regeneration method can replace ordinal TIL therapy. We added comment on this point to page 20/line 9-13 in the revised manuscript.

2. Most the TiPSC clones generated do not show improved cytotoxicity, cytokine production or persistence characteristics compared to the oligoclonal parental population. So there is no functional benefit of the rejuvenation process. The authors show cytokine production after unspecific stimulation with PMA/Ion in the main figure. However I think that the stimulation with autologous tumor should be more informative. In supplementary figure 5a none of the TiPSC-TILs shows IL-2 production after stimulation with tumor, similarly to the parental TILs. Thus, although the authors show immunophenotypic profiles of memory cells (CD62L+, CCR7+, CD28+ etc) the functional profile is far from that.

→ We thank the reviewer for the comment on the cytokine production and apoptosis sensitivity of TIL-iPS-T.

We evaluated these functions again, and the new data led us to replace some figures. In particular, all data from the apoptosis assay and some data on intracellular staining after PMA/ION stimulation were replaced. Regarding the apoptosis assay, the fractioning of each population in the previous submission seemed insufficient due to the poor performance of Annexin V-APC antibody. By using a new kit containing Annexin V-FITC and 7-AAD, we could confirm that TIL-iPS-T were more resistant to apoptosis than TI-CTL (Fig. 6d and page 16/line 2-3 in the revised manuscript). Regarding intracellular staining data with PMA/ION stimulation, we regarded the data for C-T-10 V β 2 in the previous submission inappropriate for phenotype evaluation, because the data was acquired by using TIL-iPS-T after repetitive expansion. We use data for C-T-10 V β 4 (Fig. 5b), C-T-7 V β 3 and C-T-11 V β 5.1 (Supplementary Fig. 6b) in the revised manuscript. We

comment on these findings in page 13/line 13-16.

Returning to the reviewer's concern, we agree that the data after stimulating with cancer spheroids inform us about the behavior of TIL-iPS-T within a tumor. Thus, we moved the data to Fig. 5b in the revised manuscript. However, evaluation of the CTL phenotype in the co-culture setting is difficult. Indeed, both TIL-iPS-T and the parent TI-CTL did not show IL-2 production after stimulation with the tumor, a response not always attributable to their phenotype. That is, some factors such as TCR affinity, immunogenicity of the tumor antigens, etc., affect the co-culture situation. For this reason, unspecific stimulation with PMA/ION is helpful to make the stimulating condition uniform. The relationship between phenotype and cytokine production pattern was reported with CD3 antibody plus PMA (Lipp et.al., *Nature* 1999). Thus, we left the intracellular staining data with PMA/ION stimulation as main data in Fig. 5b, while the CBA data with PMA/ION stimulation were removed.

For reference, we added new data on the relationship between the cytokine production pattern and CTL phenotype to Supplementary Fig. 6c in the revised manuscript.

3. Even the TiPSC-TIL from MMR-P samples showed improved cytotoxicity against autologous tumor spheroids. The authors in the discussion attribute that to a NK-cell activity of the TiPSC-TIL. If that is the case then why the same NK-cell activity is not observed with the non-tumor specific clones? And for the supposedly tumor-specific clones do we know which part of the cytotoxicity is NK cell-like and which is mediated through the TCR? The HLA inhibition experiment used for the selection should be repeated for the TiPSC-TILs.

→ We agree with the reviewer's comment that the non-tumor specific clones should show cytotoxicity against cancer spheroids if the additional cytotoxicity is due to NK cell-like behavior by TIL-iPS-T. To confirm this possibility, we repeatedly performed the killing assay for C-T-7 V β 5.3 in addition to another non-tumor-specific clone for C-T-10 V β 23. As the reviewer mentioned, these non-tumor specific clones exhibited killing activity against cancer spheroids to some extent. These findings are shown in Fig. 5e in the revised manuscript.

We appreciate the reviewer's recommendation to block HLA-TCR interactions during the killing assay. These additional assays indicated that TIL-iPS-T utilized both TCR-dependent and -independent killing activities, unlike TI-CTL (Supplementary Fig. 7a, b in the revised manuscript). We also checked whether TIL-iPS-T expressed NK cell-

related markers. As seen in supplementary Fig. 7c, TIL-iPS-T expressed a higher level of NK cell-related markers, especially CD336 (NKp44) and CD337 (NKp30), which are involved in non-MHC-restricted natural cytotoxicity, compared with TI-CTL. We comment on these findings in page 14/line 4-12 of the revised manuscript.

4. Since no benefit in persistence is observed compared to the parental TILs, one would expect that this at least should be overcome with the other benefit of reprogramming, the unlimited potential for dosing. However, even injection of 10 times more TiPSC-TILs did not result in a beneficial anti-tumor effect. It seems that the selected tumor model used to assess in vivo anti-tumor function is unsuccessful (since there is no positive control there) or that the generated TiPSC-TIL are actually defective in functionality. In any case there is no proof of a functional benefit through rejuvenation. In addition it is not clear which TiPSC clones were used in the in vivo experiment. Was it all 4 clones of C-T-10?

→ We apologize that we did not specify which clones were transferred in the in vivo experiments (Fig. 6 in the previous submission). We used all 4 clones of C-T-10. This information was added to supplementary Fig. 9 and page 16/line 13-14 in the revised manuscript.

We appreciate the reviewer's comment about the no apparent anti-tumor effect by the TIL-iPS-T and TI-CTL groups in the PDSX mouse model. To narrow down the possible reasons for the refractoriness, we validated the cell dynamics in this model. We applied a modified protocol which was intended to improve the persistency of the transferred cells by supplementing cytokines. The experiment revealed that TIL-iPS-T was able to accumulate at the tumor and had improved persistency compared with TI-CTL. These results about the in vivo dynamics were added to Fig. 7 and page 17/line 7-13 in the revised manuscript.

Following these experiments, we performed another experiment to evaluate the therapeutic potential of TIL-iPS-T by applying the modified protocol (see data below).

Evaluating the therapeutic potential of TIL-iPS-T by applying the modified protocol.

FF-Luc⁺ cancer spheroids were used to establish PDSX mice. All available clones (C-T-10 Vβ1, Vβ2, Vβ4 and Vβ9) were mixed at the same ratio. 7.5×10^6 TIL-iPS-T were injected 6 times. The TI-CTL group was not used in this experiment due to shortage. Cytokines (IL-2 100 U/mouse, IL-7 5 ng/mouse, IL-15 5 ng/mouse) were intraperitoneally injected three times a week.

Again, we could not obtain positive results. Because we confirmed TIL-iPS-T are competent to infiltrate the tumor, we suspect that TIL-iPS-T are not capable of inducing an adequate immune response within the tumor. Further investigation about the local behavior of TIL-iPS-T is needed.

Another possible reason for the above observation is that PDSX mouse could be an inappropriate model to demonstrate the efficacy of a cell-based immunotherapy. The model is regarded as an alternative to patient derived xenograft (PDX) models, but its utility has been reported only for sensitivity prediction to chemotherapy (Maekawa et al., *Mol. Cancer Ther.* 2018 and Yamamoto et al., *Cancers* 2020). Unlike generally used tumor cell lines, the histology of excised tumors from the PDSX model exhibited the following characteristics: aggregation of tumor cells to form a mass, growth of interstitial structures, and so on (see images below). These features could be barriers for tumor

treatment.

Subcutaneous tumor from PDSX mouse for C-T-10
(Adenocarcinoma)

Subcutaneous tumor from JHH-7 transplanted mouse
(Hepatocellular carcinoma)

Indeed, the above histological findings suggest that the PDSX model better reflects clinical features of solid tumors. However, we do not have a positive control, making it difficult to conclude whether the PDSX model works well for the evaluation of cell-based immunotherapies. Further investigation of the PDSX model are warranted. These points were added to page 20/line 16 – page 21/line 4.

Reviewer #3 (Remarks to the Author):

While reprogramming of TIL to iPSC as a method to improve ACT has been reported, the current manuscript establishes novel methods of selecting tumor-specific CTL for reprogramming using autologous cancer spheroids. The authors then go on to further establish a protocol to regenerate stable multiclonal tumor-specific TIL (TIL-iPS-T). This is a well-written and logical study that presents a careful characterization of potentially important advances on ACT methodology.

Comments:

While this study provided a logical assessment of the novel differentiation processes, I have a couple questions regarding the heterogeneity of the populations that result:

1. HLA I blocking antibody only partially inhibits TI-CTL expansion leading the authors to speculate that some of the TI-CTLs (CD56+) exhibit NK-cell like functions. Further phenotyping of the heterogeneous cell population that results from autologous cancer spheroid TI-CTL enrichment may be important to define this novel means of enrichment.

→ We thank the reviewer for the positive comment about our tumor-specific TI-CTL cloning method. Following the reviewer's suggestion, we evaluated the relationship between the inhibition efficiency of HLA-class I blocking TI-CTL activation and NK cell-related markers. We show the data below.

Markers on CD107a⁺ bulk tumor-reactive TI-CTL with or without HLA-class I blocking.

Markers on 4-1BB⁺ bulk tumor-reactive TI-CTL with or without HLA-class I blocking.

These markers were quantified by flow cytometry. Paired t-test was used.

Indeed, we narrowed down possible markers involved in the TCR-independent activation of TI-CTL to CD56, CD158e1, CD226 and CD161. However, the data were acquired from bulk tumor-reactive TI-CTL. Thus, we performed the same assays by using tumor-specific TI-CTL clones (C-T-10 Vβ1, Vβ4 and Vβ9).

Markers on tumor-reactive TI-CTL clones with or without HLA-class I blocking.

Dots represent individual data from each clone. Paired t-test was used.

As the above figure indicates, we could not find an obvious relationship between these markers and TCR-independent activation. Based on these results, we removed remarks from the revised manuscript claiming that NK cell-like functions of TI-CTL were involved in the residual activation after co-culturing with cancer spheroids in the presence of HLA-class I antibody.

Because we confirmed that repetitive bulk expansion of tumor-specific TI-CTL leads to a narrower diversity, we noted the main reason for the necessity of tumor-specific TI-CTL cloning. These results were added to supplementary Fig. 2 and page 9/ line 14-17 in the revised manuscript.

2. The TIL-iPS-T seem to differ depending on tumor source. Does this protocol result in a different population of T cells depending on the source? In other words, does the degree of differentiation of the T cells population vary? Can you get a population of T cells that are more memory-like (or stem-like; TCF+) that may be able to proliferate more or have increased SRC. A phenotyping of the T cell differentiation states within the populations of TIL-iPS-T may be important to evaluate this.

→ We would like to thank the reviewer for the comment. The quality of regenerated T

cells and the differentiation efficiency seem to mainly depend on both the protocols and the quality of the iPSC subclone. So far, we have applied feeder-free protocols for T cell differentiation to iPSC established from fibroblasts, monocytes and T cells from peripheral blood. In our experience, there is no difference in T cell differentiation efficiency or quality among these different iPSC sources. On the other hand, we found that each iPSC subclone exhibits different outcomes. In other words, there is a difference between each picked up iPSC subclone regarding T cell differentiation potential, despite the subclones having been established from the same source. It should be noted that this study deals only with colorectal cancer is the first to achieve TIL regeneration. Thus, we are not able to answer whether the TIL regeneration efficiency varies between solid tumor types. We comment on the versatility and applicability to other malignancies in the Limitations section of the revised manuscript (page 20/line 13-16).

Following the reviewer's suggestion, we checked for memory phenotype-related markers and characteristics on TIL-iPS-T. The data indicated that TIL-iPS-T upregulated TCF-1 (TCF-7) and possessed enlarged mitochondria compared with the parent TI-CTL. These findings were added to Fig. 3d, e and page 11/ line 12-15 in the revised manuscript. We could obtain TIL-iPS-T with memory-like phenotype only before expansion, so it is difficult to prepare a large number of less differentiated TIL-iPS-T for SRC evaluation. Instead, we added data about the mitochondria enlargement. It is well known that T cells with central-memory phenotype show high SRC and enlarged mitochondria biomass. We attach the data below.

CD45RA⁻CD62L⁺ CTL, regarded as a central-memory population, indicated high SRC and enlarged mitochondria biomass.

Minor point:

- The authors evaluate the number of TI-CTL in a 5mm³ tumor block and conclude based on low number that no expansion is taking place in the tumor (page 7/line 15-16). While probably that most expansion takes place before tumor infiltration, I am not sure that tumor localized expansion can be ruled out solely based on numbers.

→ We apologize for the confusion. We replaced “occurred” with “was needed” in page 7/line 15. We simply want to state the reason for the necessity of the TI-CTL expansion process before the tumor-specific TI-CTL selection phase.

Revised text:

All changes are specified with red characters in the revised manuscript. Please allow us to note only major changes in this reply document, because many changes in the revised manuscript were made.

Additions or modifications in the revised manuscripts

- Page 7/line 15; 'occurred' was replaced to 'was needed' as the response to the minor point from reviewer #3.
- Page 9/line 10 – page 10/line 3; the reasons were noted in the reply to comment 1 from reviewer #3.
- Page 10/line 17 – page 11/ line 2; the reason was noted in the reply to comment 1 from reviewer #1.
- Page 11/line 12-15; the reasons were noted in the reply to comment 2 from reviewer #3.
- Page 13/line 4-5; we added a sentence in order to specify that the subsequent data were acquired by using TIL-iPS-T after PHA/PBMC expansion.
- Page 13/line 7-9; we changed the sentence because new data were acquired from C-T-11.
- Page 13/line 10-16; the reasons were noted in the reply to comment 2 from reviewer #1.
- Page 14/line 1-12; the reasons were noted in the reply to comment 3 from reviewer #1.
- Page 14/line 17 – page 15/line 3; the summary of the paragraph was changed due to new data.
- Page 15/line 8-10; we changed the sentence because new data were acquired from C-T-11.
- Page 16/line 1-3; the reason was noted in the reply to comment 2 from reviewer #1.
- Page 16/line 4-5; summary of the paragraph was changed due new data.
- Page 16/line 13-14; the reason was noted in the reply to comment 4 from reviewer #1.
- Page 17/line 7-13; the reason was noted in the reply to comment 4 from reviewer #1.
- Page 17/line 16 – page 18/line 2; we changed the sentences for brevity.
- Page 18/line 11-14; we added a sentence.
- Page 19/line 17 – page 20/line 7; we modified the paragraph following new data.
- Page 20/line 9 – page 21/line 4; we changed the paragraph in response to comments

1 and 4 from reviewer #1 and comment 2 from reviewer #3.

Removals from the previous submission

- Page 15/line 16 – page 16/line 4 in the previous manuscript; new experiments resulted in different persistency data for PDSX mouse.
- Page 19/line 8 – page 20/line 4; we removed the paragraph for brevity.

Updated figures and tables:

The number of figures and supplementary figures were changed (6 figures and 7 supplementary figures in the previous manuscript, 7 figures and 9 supplementary figures in the revised one). All of the changes are noted below.

Additions or modifications in the revised manuscript

Fig. 1e

We added 'cytokine' to the graph labels.

Fig. 3c

The data were shown in Supplementary Fig. 3b of the previous submission.

Fig. 3d, e

These data were added following comment 2 from reviewer #3.

Fig. 4

The data for a and b were made by combining Fig. 3c and Supplementary Fig. 4a, b in the previous submission. The data for c was moved from Fig. 3d in the previous manuscript.

Fig. 5a
Proliferation data for C-T-11 were added.

Fig. 5b
The data under PMA/ION stimulation were added into Fig. 5b. The plot pattern of the PMA/ION condition was changed from pseudocolor to contour plot with outliers. As we explain in the reply to comment 2 from reviewer #1, the data for C-T-10 Vβ2 were exchanged with data for C-T-10 Vβ4.

Fig. 5c
The killing assay data for C-T-10 Vβ4 was added.

Fig. 5e

New data for C-T-10 Vβ4 and C-T-7 Vβ5.3 are shown. These changes are explained in the reply to comment 3 from reviewer #1.

Fig. 6a

Data for C-T-11 were added.

Fig. 6b

The SRC data for C-T-7 and C-T-11 were moved from Supplementary Fig. 6 in the previous manuscript to Fig. 6b in the revised one, and new data for C-T-11 Vβ20 were added. The OCR trace for C-T-10 Vβ2 was also added.

Fig. 6c, d

The order of the CFSE dilution assay data and apoptosis sensitivity data was exchanged. As we explained in the reply to comment 2 from reviewer #1, the apoptosis sensitivity data were replaced.

Fig. 7

All data are new in response to comment 4 from reviewer #1.

Supplementary Fig. 2

All data are new in response to comment 1 from reviewer #3.

Supplementary Fig. 5b

The color and pattern of the bars were changed.

Supplementary Fig. 6a, b

New data were added in response to comment 2 from reviewer #1.

Supplementary Fig. 7

New data were added in response to comment 3 from reviewer #1.

Supplementary Fig. 8b
Data on day 33 were added.

Case	Clone	Reprogramming from TI-CTL		T cell regeneration
		Colony number	Colony forming rate (%)	
C-T-7	Vβ3	462	4.62	Success
	Vβ5.3*	TNTC**		Success
	Vβ22	281	2.81	Fail
C-T-10	Vβ1	312	3.12	Success
	Vβ2	145	1.45	Success
	Vβ4	239	2.39	Success
	Vβ9	17	0.17	Success
C-T-11	Vβ23*	54	0.54	Success
	Vβ5.1	59	0.59	Success
	Vβ20	105	1.05	Success

The starting number of TI-CTL was 10,000 cells in all experiments.

* Tumor non-specific clones

** TNTC: too numerous to count

Supplementary Table. 2

The column for T cell regeneration was added in response to comment 1 from reviewer #1.

Removals from the previous submission

Fig. 4c in the previous submission

As we explained in the reply to comment 2 from t reviewer #1, the cytokine production data under PMA/ION stimulation were removed.

Fig. 5e in the previous submission

The persistency data in PDSX mouse were replaced.

Supplementary Fig. 2a in the previous submission

The reasons for the removal are explained in the reply to comment 1 from reviewer #3.

REVIEWERS' COMMENTS:

Reviewer #1 (Remarks to the Author):

The authors have adequately responded to my comments and provided the requested data. The manuscript is improved. However, from the data it seems that although this study is successful in showing the feasibility of making iPSC from TIL and re-differentiate them to produce TIL-iPSC-T there are still problems that make the application of such approach not favorable: The heterogeneity of the TILs cannot be reproduced, the TIL-iPSC-T may persist longer but it seems that they are far from efficient in anti-tumor function. The authors list now all these limitations in the discussion. They conclude that: "Our findings imply that iPSC technology has great potential for pushing TIL-ACT to the next stage". I think that it would be useful to add a paragraph with potential solutions to these limitations by further research, because if there is no way to improve these limitations then the potential for this strategy to replace TIL therapy is actually very low.

Reviewer #3 (Remarks to the Author):

This paper describes a novel methodology for selecting tumor-specific CTL for reprogramming to iPSC and subsequent regeneration of multiclonal TIL-iPS-T. The authors present a well thought-out process and a careful description of the resultant cell populations. In particular, in this version, the authors have addressed concerns by including several new pieces of data that describe the phenotypic and functional characteristics of the cell populations. Despite having minimal therapeutic effect (even with a modified protocol) in vivo in the selected model, the TIL-iPS-T cells did show some functional enhancement over TI-CTL in vitro. Therefore, I feel this manuscript will be of interest to the community and the wider field. While perhaps not ready for direct translation to clinic, this manuscript presents a methodology that warrants further discussion and experimentation by the field and so I feel merits publication.